# Leveraging Skills from Unlabeled Prior Data for Efficient Online Exploration

Max Wilcoxson [* 1]   Qiyang Li [* 1]   Kevin Frans [1]   Sergey Levine [1]

## Abstract

Unsupervised pretraining has been transformative in many supervised domains. However, applying such ideas to reinforcement learning (RL) presents a unique challenge in that fine-tuning does not involve mimicking task-specific data, but rather *exploring* and locating the solution through iterative self-improvement. In this work, we study how unlabeled offline trajectory data can be leveraged to learn efficient exploration strategies. While prior data can be used to pretrain a set of low-level skills, or as additional off-policy data for online RL, it has been unclear how to combine these ideas effectively for online exploration. Our method **SUPE** (**S**kills from **U**nlabeled **P**rior data for **E**xploration) demonstrates that a careful combination of these ideas compounds their benefits. Our method first extracts low-level skills using a variational autoencoder (VAE), and then *pseudo-labels* unlabeled trajectories with optimistic rewards and high-level action labels, transforming prior data into high-level, task-relevant examples that encourage novelty-seeking behavior. Finally, **SUPE** uses these transformed examples as additional off-policy data for online RL to learn a high-level policy that composes pretrained low-level skills to explore efficiently. In our experiments, **SUPE** consistently outperforms prior strategies across a suite of 42 long-horizon, sparse-reward tasks. Code: `https://github.com/rail-berkeley/supe`.

## 1. Introduction

Unsupervised pretraining has been transformative in many supervised domains, such as language (Devlin et al.,

2018) and vision (He et al., 2022). Pretrained models can be adapted efficiently to new tasks with few examples, and often generalize better than models trained from scratch (Radford et al., 2019; Brown et al., 2020). However, in contrast to supervised learning, reinforcement learning (RL) presents a unique challenge in that fine-tuning does not involve further mimicking task-specific data, but rather *exploring* and finding the solution through iterative self-improvement. Thus, the key challenge to address in pretraining for RL is not simply to learn good representations, but to learn an effective *exploration strategy* for solving downstream tasks.

Pretraining benefits greatly from the breadth of the data. Unlabeled trajectories (i.e., those collected from previous policies whose objectives are unknown, or from task-agnostic data collection policies) are the most abundantly available, but using them to solve specific tasks can be difficult. It is not enough to simply copy behaviors, which can differ greatly from the current task. There is an *entanglement* problem – general knowledge of the environment is mixed in with task-specific behaviors. A concrete example is learning from unlabeled locomotion behavior: we wish to learn how to move around the world, but not necessarily to the locations present in the pretraining data. We will revisit this setting in the experimental section.

The entanglement problem can be alleviated through hierarchical decomposition. Specifically, trajectories can be broken into segments of task-agnostic skills, which are composed in various ways to solve various objectives. We posit that unlabeled trajectories thus present a twofold benefit, (1) as a way to learn a diverse set of skills, and (2) as off-policy examples of composing such skills. Notably, prior online RL methods that leverage pretrained skills ignore the second benefit, and discard the offline trajectories after the skills are learned (Ajay et al., 2021; Pertsch et al., 2021; Hu et al., 2023; Chen et al., 2024). We instead argue that such trajectories are critical, and can greatly speed up learning. We make use of a simple strategy of learning an optimistic reward model from online samples, and *pseudo-labeling* past trajectories with an optimistic reward estimate. The past trajectories can thus be readily utilized as off-policy data, allowing for quick learning even with a very small number of online interactions.

---

*Equal contribution [1]University of California, Berkeley. Correspondence to: Max Wilcoxson <mwilcoxson@berkeley.edu>, Qiyang Li <qcli@eecs.berkeley.edu>.

*Proceedings of the 42nd International Conference on Machine Learning*, Vancouver, Canada. PMLR 267, 2025. Copyright 2025 by the author(s).

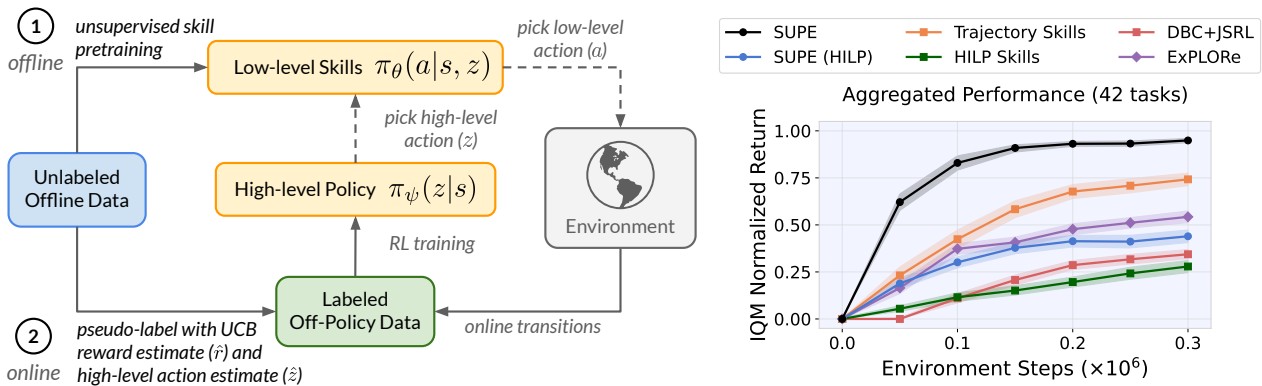

Figure 1: **Left:** **SUPE** utilizes unlabeled offline data *twice*, both for *offline* unsupervised skill pretraining and *online* RL for learning the high-level policy. We *pseudo-label* the unlabeled data with upper-confidence bound (UCB) estimates of the true rewards (*reward label*: $\hat{r}$) and high-level action estimates (*high-level action label*: $\hat{z}$). This transforms the unlabeled data into high-level labeled off-policy data that encourages *exploration*, allowing significantly more sample efficient learning online. **Right:** Empirically, **SUPE** attains strong performance across 42 challenging sparse reward tasks (see details in Section 5).

We formalize these insights as **SUPE** (**S**kills from **U**nlabeled **P**rior data for **E**xploration), a recipe for maximally leveraging unlabeled offline data for online exploration (Figure 1). The offline data is utilized during both the offline and online phases. ①  In the offline phase, we extract short segments of trajectories from the unlabeled offline data and use them to pretrain a space of low-level skills represented by a latent conditioned policy $\pi_\theta(a|s, z)$. ②  In the online phase, we utilize the unlabeled offline data as additional off-policy data for learning a high-level exploration policy $\pi_\psi(z|s)$ by *pseudo-labeling* each trajectory segment with an upper-confidence bound (UCB) reward estimate of the true rewards and a high-level action estimate ($\hat{z}$). By using offline data in these two ways, we utilize both its low and high-level structure to accelerate online exploration.

Our main contribution is a simple method that leverages unlabeled offline trajectory data to both pretrain skills offline and compose these skills for efficient online exploration. We instantiate **SUPE** with a variational autoencoder (VAE) to extract low-level skills, and an off-the-shelf off-policy RL algorithm (Ball et al., 2023) to learn a high-level policy from both online and offline data. Our empirical evaluations on a set of challenging sparse reward tasks show that leveraging the unlabeled offline data during both offline and online learning is crucial for efficient exploration, enabling **SUPE** to find sparse reward signals more quickly and achieve more efficient learning compared to all prior methods (none of which utilize the offline data during both online and offline learning).

## 2. Related Work

**Unsupervised skill discovery** methods were first investigated in the online setting, where RL agents were tasked with learning structured behaviors in the absence of a reward signal (Gregor et al., 2016; Bacon et al., 2017; Florensa et al., 2017; Achiam et al., 2018; Eysenbach et al., 2018; Sharma et al., 2020; Hansen et al., 2020; Campos et al., 2020; Laskin et al., 2021; 2022; Park et al., 2023b; Yang et al., 2023; Bai et al., 2024). These insights naturally transferred to the offline setting as a method of pretraining on unlabeled trajectory data. Offline skill discovery methods largely comprise of two categories, those who extract skills based on optimizing unsupervised reward signals (in either the form of policies (Touati et al., 2022; Hu et al., 2023; Frans et al., 2024; Park et al., 2024b) or Q-functions (Chen et al., 2024)), and those who utilize conditional behavior-cloning over subsets of trajectories (Paraschos et al., 2013; Merel et al., 2018; Shankar & Gupta, 2020; Ajay et al., 2021; Singh et al., 2021; Pertsch et al., 2021; Nasiriany et al., 2022). Closest to our method are Ajay et al. (2021) and Pertsch et al. (2021), who utilize a trajectory-segment VAE to pretrain low-level skills offline and learn a high-level policy online to compose them. In contrast to these methods that utilize offline data purely for offline skill-learning, our method also utilizes the data during online learning via *pseudo-labeling*. As we show in our experiments, utilizing both the low-level and high-level structure of the offline trajectories is critical for fast exploration and learning online.

**Offline-to-online reinforcement learning** methods (Xie et al., 2021b; Song et al., 2023; Lee et al., 2022; Agarwal et al., 2022; Zhang et al., 2023; Zheng et al., 2023; Ball et al., 2023; Nakamoto et al., 2024) focus on using task-specific offline data to accelerate online learning. Many

offline RL approaches can be applied to this setting — simply run offline RL first on the offline data to convergence as an initialization and then continue training for online learning (using the combined dataset that consists of both offline and online data) (Kumar et al., 2020; Kostrikov et al., 2021; Tarasov et al., 2024). Such approaches often result in slow online improvements as offline RL objectives tend to overly constrain the policy behaviors to be close to the offline data, impairing exploration. On the other hand, off-policy online RL methods can also be directly applied in this setting by treating the offline data as additional off-policy data in the replay buffer and learning the policy from scratch (Lee et al., 2022; Song et al., 2023; Ball et al., 2023). While related in spirit, these methods cannot be directly used in our setting as they *require* offline data to have reward labels.

**Data-driven exploration.** A common approach for online exploration is to augment rewards with a bonus that incentives exploration and optimize the agent with respect to these modified rewards (Stadie et al., 2015; Bellemare et al., 2016; Houthooft et al., 2016; Pathak et al., 2017; Tang et al., 2017; Ostrovski et al., 2017; Achiam & Sastry, 2017; Merel et al., 2018; Burda et al., 2018; Ermolov & Sebe, 2020; Guo et al., 2022; Lobel et al., 2023). While most exploration methods operate in the purely online setting and focus on adding bonuses to the online replay buffer, recent works have begun to explore a more data-driven approach that makes use of unlabeled offline data to guide online exploration. Li et al. (2024) added bonuses to the offline data, incentivizing exploration around the offline data distribution. While prior methods apply the bonus directly to dataset transitions, we instead add bonuses to high-level transitions, allowing us to learn a policy that composes pretrained skills effectively for exploration. Hu et al. (2023) proposes a different strategy, first learning a number of policies using offline RL that each optimizes for a random reward function, then sampling from these policies online to achieve structured exploratory behavior. This approach does not utilize offline data during the online phase and requires each policy corresponding to a different reward function to be represented separately. In contrast, our method makes use of the offline data as off-policy data for updating the high-level policy and our skills are represented using a single network parameterized by a skill latent. As we will show, using offline data online is crucial for efficient exploration.

**Hierarchical reinforcement learning (HRL).** The ability of RL agents to solve long-horizon exploration tasks is an important research goal in the field of HRL (Dayan & Hinton, 1992; Dietterich, 2000; Vezhnevets et al., 2016; Daniel et al., 2016; Kulkarni et al., 2016; Vezhnevets et al., 2017; Peng et al., 2017; Riedmiller et al., 2018; Nachum et al., 2018; Ajay et al., 2021; Shankar & Gupta, 2020; Pertsch et al., 2021; Gehring et al., 2021; Xie et al., 2021a).

HRL methods typically learn a high-level policy to leverage a space of low-level primitive policies online. These primitives can be either manually specified (Dalal et al., 2021), learned online (Dietterich, 2000; Kulkarni et al., 2016; Vezhnevets et al., 2016; 2017; Nachum et al., 2018), or pretrained using unsupervised skill discovery methods as discussed above. While many existing works learn or fine-tune the primitives online along with the high-level policy (Dietterich, 2000; Kulkarni et al., 2016; Vezhnevets et al., 2016; 2017; Nachum et al., 2018; Shankar & Gupta, 2020), others opt for a simpler formulation where the primitives are fixed after an initial pre-training phase and only the high-level policy is learned online (Peng et al., 2017; Riedmiller et al., 2018; Ajay et al., 2021; Pertsch et al., 2021; Gehring et al., 2021). Our work adopts the later strategy where we pre-train skills offline using a static, unlabeled dataset. No prior HRL method simultaneously leverages offline data for skill pre-training and as additional off-policy data for learning a high-level policy online. As we show in our experiments, using these approaches in combination is crucial to achieving sample efficient learning on challenging sparse-reward tasks.

See more related work discussions on options and unsupervised pre-training in Appendix D.

## 3. Problem Formulation

In this paper, we consider a Markov decision process (MDP) $\mathcal{M} = \{\mathcal{S}, \mathcal{A}, \boldsymbol{P}, \gamma, r, \rho\}$ where $\mathcal{S}$ is the set of all possible states, $\mathcal{A}$ is the set of all possible actions that a policy $\pi(a|s) : \mathcal{S} \mapsto \mathcal{P}(\mathcal{A})$ may take, $\boldsymbol{P}(s'|s, a) : \mathcal{S} \times \mathcal{A} \mapsto \mathcal{P}(\mathcal{S})$ is the transition function that describes the probability distribution over the next state $s'$ given the current state and the action taken at the state, $\gamma$ is the discount factor, $r(s, a) : \mathcal{S} \times \mathcal{A} \mapsto \mathbb{R}$ is the reward function, and $\rho : \mathcal{P}(\mathcal{S})$ is the initial state distribution. We have access to a dataset of trajectories that are collected from the same MDP with no reward labels: $\mathcal{D} := \{\tau^1, \tau^2, \cdots\}$ (where $\tau^i := \{s_0^i, a_0^i, s_1^i, a_1^i, s_2^i, \cdots\}$). During online learning, the agent may interact with the environment by taking actions and observes the next state and the reward specified by transition function $\mathcal{P}$ and the reward function $r$. We aim to develop a method that can leverage the dataset $\mathcal{D}$ to efficiently explore in the MDP to collect reward information, and outputs a well-performing policy $\pi(a|s)$ that achieves high cumulative return in the environment $\eta(\pi) := \mathbb{E}_{\{s_0 \sim \rho, a_t \sim \pi(a_t|s_t), s_{t+1} \sim \boldsymbol{P}(\cdot|s_t, a_t)\}} \sum_{t=0}^{\infty} [\gamma^t r(s_t, a_t)]$. Note that this is different from the zero-shot RL setting (Touati et al., 2022) where the reward function is specified for the online evaluation (only unknown during the unsupervised pretraining phase). In our setting, the agent has no knowledge of the reward function a priori

**Algorithm 1 SUPE**

**Input:** Unlabeled dataset of trajectory segments $\mathcal{D}$, segment length $H$, batch size $B$, replay buffer $\mathcal{D}_{\text{replay}} \leftarrow \emptyset$.

**1. Unsupervised Skill Pretraining**

**for** each pretraining step **do**
$\quad \{\tau^1_{[H]}, \cdots, \tau^B_{[H]}\} \sim \mathcal{D}$
$\quad$ Update $\theta$ with $\nabla_\theta \frac{1}{B} \sum_{i=1}^B \mathcal{L}_\theta(\tau_i)$ (Equation 1)

**2. Online RL with Pseudo-labeling**

**for** every $H$ online environment steps **do**

$\quad$ **2a. Environment Interaction**
$\quad z \sim \pi_\psi(z|s)$
$\quad$ Run $\pi_\theta(a|s,z)$ for $H$ steps: $\{s_0, a_0, r_0, \cdots, s_H\}$
$\quad \mathcal{D}_{\text{replay}} \leftarrow \mathcal{D}_{\text{replay}} \cup \{(s_0, z, s_H, \sum_{i=0}^{H-1}[\gamma^i r_i])\}$.

$\quad$ **2b. Pseudo-labeling Offline Data**
$\quad \{\tau^1_{[H]}, \cdots, \tau^B_{[H]}\} \sim \mathcal{D}$ ▷ *traj. segments of length H*
$\quad \hat{z}^i \sim f_\theta(z|\tau^i_{[H]}), \forall i$ ▷ *high-level action estimate*
$\quad \hat{r}^i \leftarrow r_{\text{UCB}}(s_0^i, \hat{z}^i), \forall i$ ▷ *UCB reward estimate*
$\quad \boldsymbol{B}_{\text{offline}} \leftarrow \{(s_0^i, \hat{z}^i, \hat{r}^i, s_H^i)\}_{i=1}^B$ ▷ *assemble batch*

$\quad$ **2c. Updating High-level Policy** $\pi_\psi$
$\quad \boldsymbol{B}_{\text{online}} \sim \mathcal{D}_{\text{replay}}$
$\quad$ Update $\psi$ with off-policy RL on $\boldsymbol{B}_{\text{online}} \cup \boldsymbol{B}_{\text{offline}}$

**Output:** A hierarchical policy consisting of a high-level $\pi_\psi(z|s)$ and low-level $\pi_\theta(a|s,z)$

---

and must actively explore the environment online to receive reward information. In addition, our setting is also different from an adjacent setting where offline data only contain observations (Ma et al., 2022; Ghosh et al., 2023; Song et al., 2024) (see Appendix D for more discussions).

# 4. Skills from Unlabeled Prior Data for Exploration (SUPE)

In this section, we describe how we utilize the unlabeled trajectory dataset to accelerate online exploration. Our method, **SUPE**, can be divided into two parts. The first part is the offline pretraining phase where we extract skills from the offline data with a trajectory-segment VAE. The second part is the online learning phase where we train a high-level off-policy agent to compose the pretrained skills by leveraging examples from both the offline data and online replay buffer. Algorithm 1 describes our method.

**Pretraining with trajectory VAE.** Without knowledge of the downstream task, we need to capture the diverse range of multitask behavior from the unlabeled offline dataset as accurately as possible. At the same time, we aim to leverage this dataset to more efficiently learn a high-level policy which composes these skills to solve a par-

ticular unknown task online. We achieve this by adopting a trajectory VAE design from prior methods (Ajay et al., 2021; Pertsch et al., 2021) where a short trajectory segment $\tau_{[H]} = \{s_0, a_0, s_1, \cdots, s_{H-1}, a_{H-1}\}$ is first fed into a trajectory encoder $f_\theta(z|\tau)$ that outputs a distribution over the skill latent $z$. Then, a skill policy $\pi_\theta(a|s,z)$ is trained to reconstruct the actions in the segment using a latent sampled from the distribution outputted by the encoder. This design allows us to use the trajectory encoder $f_\theta(z|\tau)$ to obtain an estimate of the corresponding skill latent $\hat{z}$ for any trajectory segment. Following prior work (Pertsch et al., 2021), we learn a state-dependent prior $p_\theta(z|s)$ to help accommodate the difference in behavior diversity of different states. Putting them all together, the loss function is

$$\mathcal{L}_\theta(\tau) = \beta D_{\text{KL}}(f_\theta(z|\tau)||p_\theta(z|s_0)) - \\ \mathbb{E}_{z \sim f_\theta(z|\tau)} \left[ \sum_{h=0}^{H-1} \log \pi_\theta(a_h|s_h, z) \right]. \quad (1)$$

**Online exploration with trajectory skills.** To effectively leverage the pretrained trajectory skills, we learn a high-level off-policy agent that picks which skills to use to explore and solve the task efficiently. Following prior work on fixed-horizon skills (Pertsch et al., 2021), our high-level policy selects a latent skill $z$ every $H$ environment steps. To transform the unlabeled offline data into high-level, task-relevant transitions, we need both the high-level action and the reward label corresponding to each length $H$ trajectory segment $\tau_{[H]}$. For the high-level action, we use the trajectory encoder $f_\theta(z|\tau)$ to obtain an estimate of the skill latent $\hat{z}$. For the reward, we maintain an upper-confidence bound (UCB) estimate of the reward value for each state and skill latent pair $(s, z)$ inspired by prior work (Li et al., 2024) (where it does so directly in the state-action space $(s, a)$), and pseudo-label the transition with such an optimistic reward estimate. The reward estimate is recomputed before updating the high-level agent, since the estimate changes over time, while the trajectory encoding is computed before starting online learning, since this label does not change. The *pseudo-labeling* is summarized below:

$$(\underbrace{s_0}_{\text{state}}, \underbrace{\hat{z} \sim f_\theta(z|\tau)}_{\text{labeled action}}, \underbrace{\hat{r} = r_{\text{UCB}}(s_0, \hat{z})}_{\text{labeled reward}}, \underbrace{s_H}_{\text{next state}}). \quad (2)$$

**Practical implementation details.** Following prior work on trajectory-segment VAEs (Ajay et al., 2021; Pertsch et al., 2021), we use a Gaussian distribution (where both the mean and diagonal covariance are learnable) for the trajectory encoder, the skill policy, as well as the state-dependent prior. While Pertsch et al. (2021) use a KL constraint between the high-level policy and the state-dependent prior, we use a simpler design without the KL constraint that performs better in practice (as we show in Appendix G).

To achieve this, we adapt the policy parameterization from Haarnoja et al. (2018), where the action value is enforced to be between $-1$ and $1$ using a tanh transformation, and entropy regularization is applied on the squashed space. We use this policy parameterization for the high-level policy $\pi(z|s)$ to predict the skill action in the squashed space $z_{\text{sqaushed}}$. We then recover the actual skill action vector by unsquashing it according to $z = \text{arctanh}(z_{\text{sqaushed}})$, so that it can be used by our skill policy $\pi_\theta(a|s,z)$. For upper-confidence bound (optimistic) estimation of the reward, $(r_{\text{UCB}}(s_0, \hat{z}))$, we directly borrow the UCB estimation implementation in Li et al. (2024) (Section 3, practical implementation section in their paper), where they use a combination of the random network distillation (RND) (Burda et al., 2018) reward bonus and the predicted reward from a reward model (see Appendix C for more details). For the off-policy high-level agent, we follow Li et al. (2024) to use RLPD (Ball et al., 2023) that takes a balanced number of samples from the offline data and the online replay buffer for agent optimization. In addition to using the optimistic offline reward label, we also add the RND bonus to the online batch as we have found it to further accelerate online learning.

## 5. Experimental Results

We present a series of experiments to evaluate the effectiveness of our method in discovering fast exploration strategies. We specifically focus on long-horizon, sparse-reward settings, where online exploration is especially important. In particular, we aim to answer the following questions:

1. Can **SUPE** leverage pretrained skills to accelerate online learning?
2. Can **SUPE** find goals faster than prior methods?
3. How sensitive is **SUPE** to hyperparameters?

### 5.1. Experimental setup

We conduct our experiments on 8 challenging sparse-reward domains and provide a brief description of each of the domains below (more details available in Appendix E).

**State-based locomotion:** antmaze, humanoidmaze, antsoccer. The first set of domains involve controlling and navigating a robotic agent in a complex environment. antmaze is a standard benchmark for offline-to-online RL from D4RL (Fu et al., 2020). humanoidmaze and antsoccer are locomotion domains from OGBench, a offline goal-conditioned RL benchmark (Park et al., 2024a). In antmaze and humanoid, a humanoid or ant agent must learn to navigate to various goal locations. antsoccer adds the difficulty of moving a soccer ball to a desired position. Each of the antmaze and humanoidmaze domains has three different maze layouts. For antmaze, we test on four different goal locations for each maze layout. For humanoidmaze and antsoccer we test on one goal. For humanoidmaze, we use both the navigate and stitch datasets. For antsoccer, we use the navigate dataset for the arena and medium maze layouts.

**State-based manipulation:** kitchen, cube-single, cube-double, scene. Next, we consider a set of manipulation domains that require a wide range of manipulation skills. kitchen is a standard benchmark from D4RL where a robotic arm needs to complete a set of manipulation tasks (e.g., turn on the microwave, move the kettle) in sequence in a kitchen scene. cube-single, cube-double, and scene are three domains from OGBench (Park et al., 2024a). For both cube-* domains, the robotic arm must arrange one or more cube objects to desired goal locations (e.g., stacking on top of each other) in tasks that mainly involve composing pick and place motions. For scene, the robotic arm can interact with a more diverse set of objects: a window, a drawer, a cube and two locks that control the window and the drawer. The tasks in scene are also longer, requiring the composition of multiple atomic behaviors (e.g., locking and unlocking, opening the drawer/window, moving the cube).

**Pixel-based:** visual-antmaze is a challenging visual domain introduced by Park et al. (2023a). The agent has all the same proprioceptive information from the state-based antmaze environment except the agent's $(x, y)$ position, which the agent must obtain from a $64 \times 64$ image observation of its surroundings.

To evaluate our method on these domains, we take the datasets in these benchmarks and remove the reward label as well as any information in the transition that may reveal the information about the termination of an episode. For all of the domains above, we use the normalized return, a standard metric for D4RL (Fu et al., 2020) environments, as the main evaluation metric. For the kitchen domain, the normalized return represents the average percentage of the tasks that are solved. For tasks in other domains, the normalized return represents the average task success rate. For all our plots, the shaded area indicates the 95% stratified bootstrap confidence interval and the solid line indicates the interquartile mean following Agarwal et al. (2021). See more details about the tasks and rewards in Appendix E.

### 5.2. Comparisons

While there are no existing methods that pretrain on unlabeled offline trajectories and use these trajectories for online learning, there are methods that use offline data for either purpose. We first consider a baseline that directly performs online learning with no pretraining.

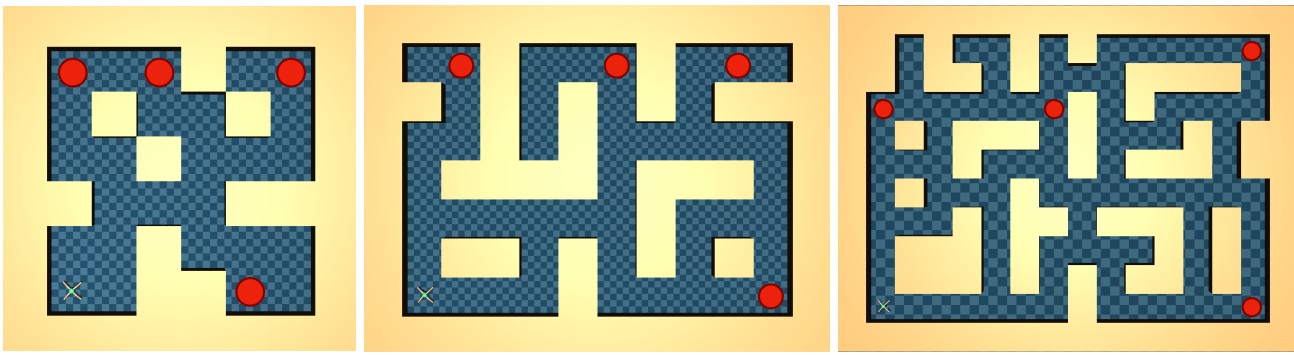

a) `antmaze`: three maze layouts (`medium`, `large` and `ultra`), and four goals for each layout.

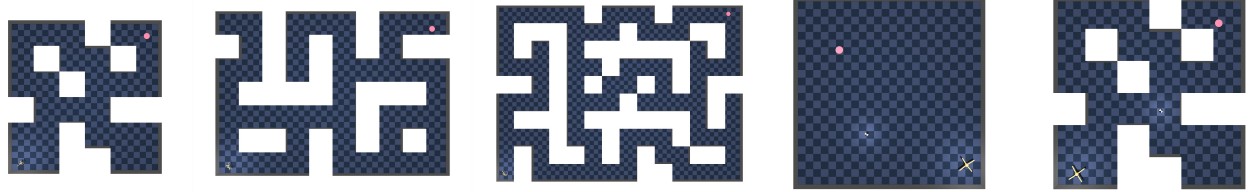

b) `humanoidmaze-medium/large/giant.`

c) `antsoccer-arena/medium`

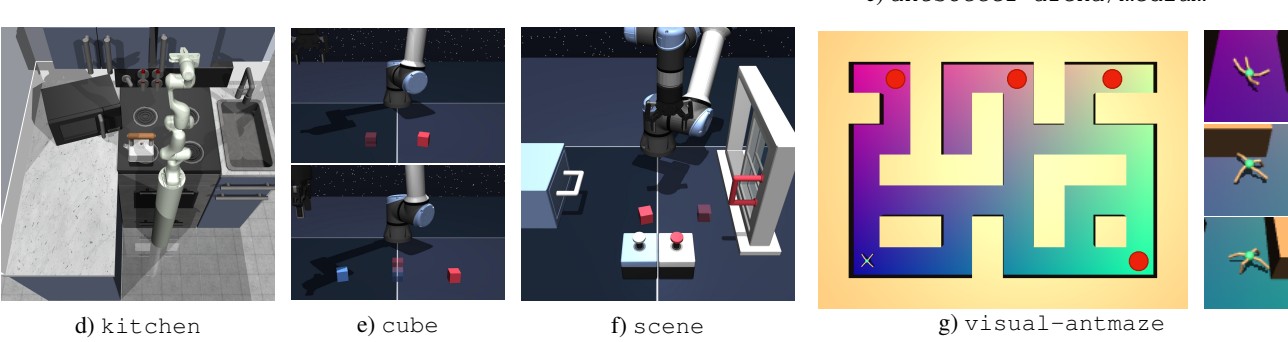

d) `kitchen`      e) `cube`      f) `scene`      g) `visual-antmaze`

Figure 2: **We experiment on 8 challenging, sparse-reward domains.** *a)*: `antmaze` with three different layouts and the corresponding four goal locations (denoted as the red dots); *b):* `humanoidmaze` with three layouts; *c)*: `antsoccer` with two layouts *d)*: `kitchen`; *e)*: `cube-single` and `cube-double`; *f)*: `scene`; *g)*: `visual-antmaze` with colored floor and local $64 \times 64$ image observations.

**EXPLORE** (Li et al., 2024). Unlike our method, this baseline does not perform unsupervised skill pretraining and directly learns a 1-step policy online. As we will show, pretraining is crucial for success on more challenging domains. Like our method, **EXPLORE** uses an exploration bonus on offline data to encourage exploration. It is worth noting that the original **EXPLORE** method does not make use of an online reward bonus and only uses the RND network to optimistically label offline data. We additionally add the reward bonus to the online batch from the replay buffer to help the agent explore beyond the offline data distribution. For completeness, we include the performance of the original **EXPLORE** in Figure 12. For all the baselines below (including our method), we add the RND bonus to the online training batch to ensure a fair comparison.

We then consider baselines that pretrain on unlabeled offline data but do not use the data online.

**DBC+JSRL**. We pretrain an expressive diffusion policy to imitate the offline data via a behavior cloning (BC) loss. At the beginning of each online episode, we rollout this pretrained policy for a random number of steps from the initial state before using switching to the online RL agent (Uchendu et al., 2023; Li et al., 2023). One might expect that a sufficiently expressive policy class can capture the action distribution of the offline data accurately enough to provide an informative initial state distribution for online exploration. This baseline is an upgraded version of the **BC+JSRL** baseline used in **EXPLORE** (Li et al., 2024), with the Gaussian BC policy replaced by a diffusion policy.

**TRAJECTORY SKILLS** and **HILP SKILLS**. We also consider two baselines that pretrain skills offline, but discard the offline data during online learning, training the high-level policy from scratch. Each baseline uses a different skill pretraining method. In addition to the trajec-

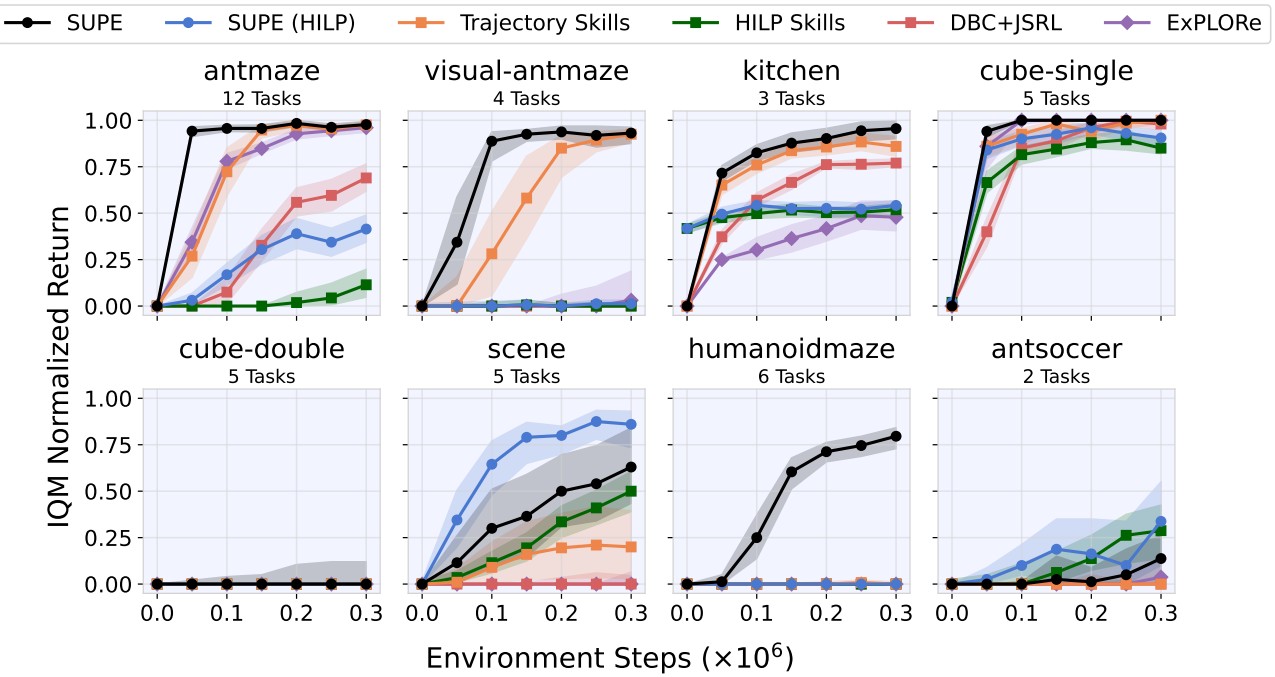

Figure 3: **Aggregated normalized return across eight different domains.** We evaluate baselines that use offline data only for pretraining (■), only during online learning (◆), and our methods that use offline data for both pretraining and online learning (●). **SUPE** matches or outperforms all other methods on all domains except `antsoccer` and `scene`, where **SUPE (HILP)** performs better. Section 5.2 contains more details about the baselines. For `kitchen`, we use 16 seeds. For all other domains, we use 8 seeds.

tory skill approach used by our method, we also consider a recently proposed unsupervised offline skill discovery method where skills are pretrained to traverse a learned Hilbert representation space (Park et al., 2024b) (HILP). For both baselines, we use the same high-level RL agent as our method, but learn this high-level policy purely from online interaction with the environment. While learning a high-level policy from scratch to compose trajectory VAE skills is similar to SPiRL (Pertsch et al., 2021), our baseline features two additional improvements. The first improvement is replacing the KL constraint with entropy regularization (same as **SUPE** as described in Section 4, practical implementation details). The second improvement is the online RND bonus that is also added to all other methods.

Finally, we introduce a novel baseline that also uses offline data during pretraining and online exploration, but uses HILP skills rather than trajectory-based skills.

**SUPE (HILP)**. Like trajectory skills, we can also estimate the corresponding HILP skill latent for any trajectory segment. HILP learns a latent space of the observations (via an encoder $\phi_{\text{HILP}}$) and learns skills that move the agent in a certain direction $z$ in the latent space. For any high-level transition $(s_0, s_H)$, we can simply take $\hat{z} \leftarrow$ normalize$(\phi_{\text{HILP}}(s_H) - \phi_{\text{HILP}}(s_0))$, the normalized difference vector that points from $s_0$ to $s_H$ in the latent space.

Aside from skill labeling, we use the exact same high-level RL agent as used in our method.

For `visual-antmaze`, we use the same image encoder used in RLPD (Ball et al., 2023). We also follow one of our baselines, **EXPLORE** (Li et al., 2024), to use ICVF (Ghosh et al., 2023), a method that uses task-agnostic value functions to learn image/state representations from passive data. ICVF takes in an offline unlabeled trajectory dataset with image observations and pretrain an image encoder in an unsupervised manner. Following **EXPLORE**, we take the weights of the image encoder from ICVF pretraining to initialize the image encoder's weights in the RND network. To make the comparison fair, we also apply ICVF to all baselines (details in Appendix F).

### 5.3. Can SUPE leverage pretrained skills to accelerate online learning?

Figure 3 shows the aggregated performance of our approach on all eight domains. Our method outperforms all prior methods on all domains except `antsoccer`, where **HILP SKILLS** performs better, and `cube-single`, where **EXPLORE** performs competitively. Additionally, we note that while **SUPE (HILP)** performs best on `scene` and `antsoccer`, **SUPE** performs best on all other tasks. This suggests that different skill pretraining

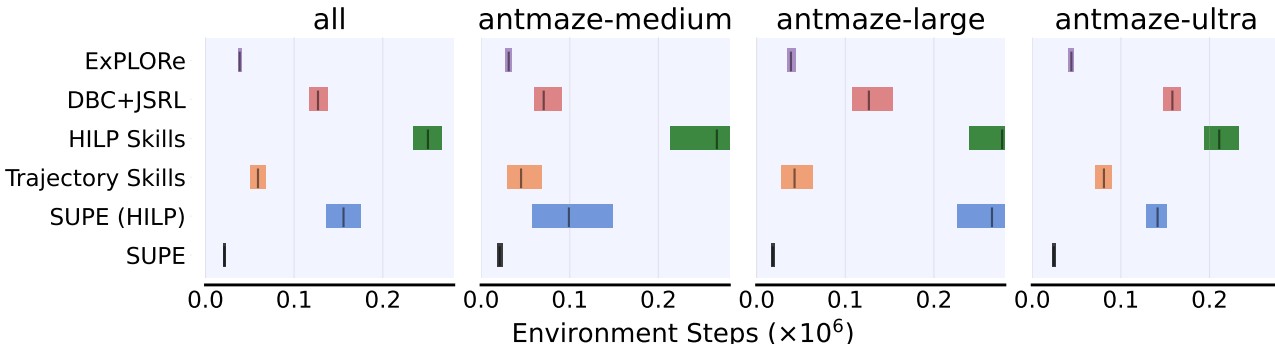

Figure 4: **Interquartile mean (IQM) of the number of environment steps taken to reach the goal (lower the better).** The first goal time is considered to be the maximum ($0.3 \times 10^6$ steps) if the agent never finds the goal. **SUPE** is the most consistent, achieving performance better than all other baselines on all layouts (4 tasks/goals for each layout and 8 seeds for each task).

.

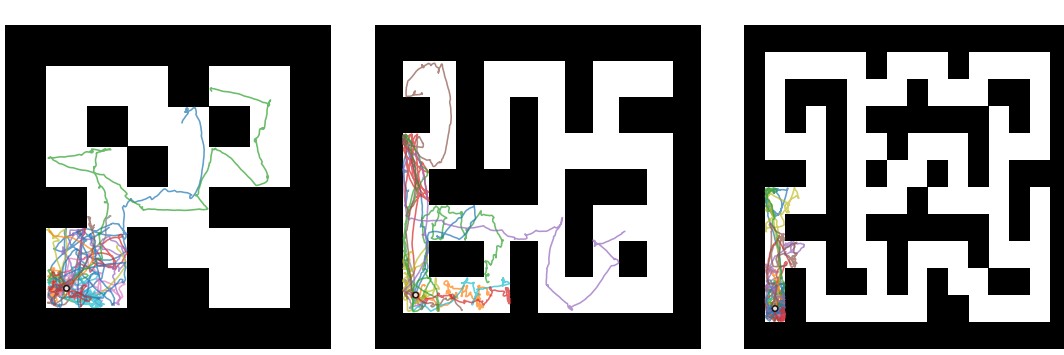

Figure 5: **SUPE skill policy rollouts in `humanoid-maze-(medium/large/giant)`.** For each trajectory, we first sample a skill latent, and then repeatedly sample a lower-level action from the low-level skill policy conditioned on that fixed latent. We plot the $x$-$y$ position of the agent throughout each of 16 such trajectories for each maze.

methods may excel at different tasks. Unlike all prior methods, both **SUPE** and **SUPE (HILP)** leverage offline data twice (both during offline pretraining and online learning), which demonstrates how this new approach is critical to surpassing prior methods. Both skill-based online learning baselines (**TRAJECTORY SKILLS** and **HILP SKILLS**) each consistently perform worse than the corresponding novel method that uses offline data during online learning (**SUPE** and **SUPE (HILP)**), which further demonstrates the importance of using offline data twice. **EXPLORE** uses offline data during online learning, but does not pretrain skills, leading to slower learning on all 8 domains and difficulty achieving any significant return on any domains other than the easier `antmaze` and `cube-single` tasks. We also report the performance on individual `antmaze` mazes and `kitchen` tasks in Figure 11, and observe that our method outperforms the baselines more on harder environments. This trend continues with `humanoidmaze`, where **SUPE** is the only method to achieve nonzero final return on the more difficult large and giant mazes (see Appendix J.1, Figure 16). These experiments suggest that pretraining skills from prior data and leveraging offline data in

the online phase are *both* crucial for sample-efficient learning, especially in more challenging environments.

To provide more insight on the sample-efficiency of our method, we also provide a visualization of rollouts from the skill policy in Figure 5. We notice that some of the frozen skill policies are able to navigate quite far from the starting point, which suggests that such structured navigation behaviors allow the high-level policy to effectively explore with a reduced time horizon.

For the visual domain, we perform an ablation study to assess the importance of the ICVF pretrained representation, which we include in Appendix F. While ICVF combines synergistically with our method to further accelerate learning, initializing RND image encoder weights using ICVF is not critical to its success.

### 5.4. Can SUPE find goals faster than prior methods?

While we have demonstrated that our method outperforms prior methods, it is still unclear if our method actually leads to better exploration (instead of simply learning the high-

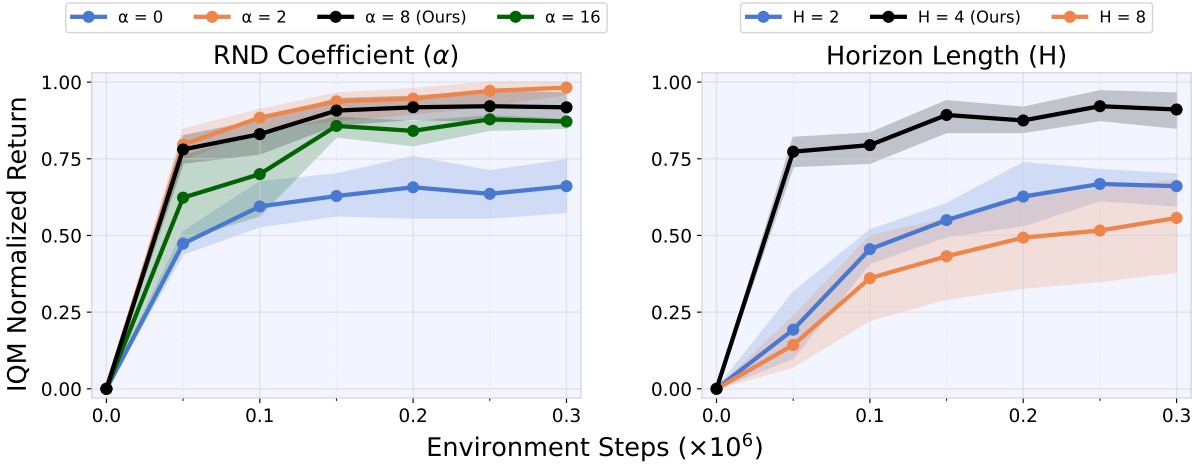

Figure 6: **Sensitivity analysis on the RND coefficient ($\alpha$) and the skill horizon length ($H$) on a subset of tasks.** For different hyperparameter values, we report the interquartile mean (IQM) of the normalized return across seven representative tasks. The performance of **SUPE** is not very sensitive to the magnitude of $\alpha$ as long as it is within a reasonable range $(2, 16)$. Without the bonus ($\alpha = 0$), **SUPE** performs significantly worse. A skill horizon length of 4 performs significantly better than a horizon length of 2 or 8. We use a skill length of 4 and $\alpha = 8$ for all experiments with skill-based methods. The normalized return for each individual task we use for this analysis can be found in Appendix H. We average over 4 seeds.

level policy better). In this section, we study the exploration aspect in isolation in the `antmaze` domain. Figure 4 reports the number of online environment interaction steps for the agent to reach the goal for the first time. Such a metric allows us to assess how efficiently the agent explores in the maze. Figure 4 shows that our method reaches goals faster than every baseline on all three maze layouts, which confirms that our method not only learns faster, but does so by exploring more efficiently. We include the results for each individual goal location and maze layout in the Appendix (Table 3) for completeness.

### 5.5. How sensitive is SUPE to hyperparameters?

Finally, we analyze the sensitivity of our method to hyperparameter selection. We focus on two hyperparameters, 1) $\alpha$: the amount of optimism (RND coefficient) used when labeling offline data with UCB rewards (Equation 2), and 2) $H$: the length of the skill. We select one representative task from each state-based domain, and study how different $\alpha$ and $H$ values affect the performance on these tasks (Figure 6). When the RND bonus is removed ($\alpha = 0$), our method performs significantly worse, highlighting the importance of optimistic labeling on efficient online exploration. While we observe some variability across individual tasks (Appendix H, Figure 9), our method is largely not very sensitive to the RND coefficient value. The aggregated performance is similar for $\alpha \in \{2, 8, 16\}$. Another key hyperparameter in our method is the skill horizon length (see Figure 6, right). We find that while there is some variability across individual tasks (Appendix H, Figure 10), a skill horizon length of 4 generally performs the best, and shorter

or longer horizons perform much worse.

## 6. Discussion and Limitations

In this work, we propose **SUPE**, a method that leverages unlabeled offline trajectory data to accelerate online exploration and learning. The key insight is to use unlabeled trajectories *twice*, to 1) extract a set of low-level skills offline, and 2) serve as additional data for a high-level off-policy RL agent to compose these skills to explore in the environment. This allows us to effectively combine the strengths from unsupervised skill pretraining and sample-efficient online RL methods to solve a series of challenging long-horizon sparse reward tasks significantly more efficiently than existing methods. Our work opens up avenues in making full use of prior data for scalable, online RL algorithms. First, our pretrained skills remain frozen during online learning, which may hinder online learning when the skills are not learned well or need to be updated as the learning progresses (see analysis in Appendix K and a failure case in Figure 19). Such problems could be alleviated by utilizing a better skill pretraining method, or allowing the low-level skills to be fine-tuned online. In addition, our approach relies on RND to maintain an upper confidence bound on the optimistic reward estimate. Although we find that RND works without ICVF on high-dimensional image observations in `visual-antmaze`, the use of RND in other high dimensional environments may require more careful consideration. Possible future directions include examining alternative methods of maintaining this bound.

## Acknowledgments

This research used the Savio computational cluster resource provided by the Berkeley Research Computing program at UC Berkeley, and was supported by ONR through N00014-22-1-2773, AFOSR FA9550-22-1-0273, and the AI Institute. We would like to thank Seohong Park for providing his implementation of OPAL which was adapted and used for skill pretraining in our method. We would also like to thank Seohong Park, Fangchen Liu, Junsu Kim, Dibya Ghosh, Katie Kang, Oleg Rybkin, Kyle Stachowicz, Zhiyuan Zhou for discussions on the method and feedback on the early draft of the paper.

## Impact Statement

This paper presents work whose goal is to advance the field of reinforcement learning. There are many potential societal consequences of our work, none of which we feel must be specifically highlighted here.

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

## A. Compute Resources

We ran all our experiments on NVIDIA A5000 GPU and V100 GPUs. In total, the results in this paper required approximately 16600 GPU hours.

## B. VAE Architecture and hyperparameters

We use a VAE implementation from Park et al. (2024b). The authors kindly shared with us their OPAL implementation (which produces the results of the OPAL baseline in the paper). In this implementation, the VAE encoder is a recurrent neural network that uses gated-recurrent units (Cho et al., 2014) (GRU). It takes in a short sequence of states and actions, and produces a probabilistic output of a latent $z$. The reconstruction policy decoder is a fully-connected network with ReLU activation (Nair & Hinton, 2010) that takes in both the state in the sequence as well as the latent $z$ to output an action distribution.

| Parameter Name | Value |
|---|---|
| Batch size | 256 |
| Optimizer | Adam |
| Learning rate | $3 \times 10^{-4}$ |
| GRU Hidden Size | 256 (`antmaze`, `kitchen`, `visual-antmaze`) |
| | 512 (`antsoccer`, `humanoidmaze`, `scene`, `cube-*`) |
| GRU Layers | 2 hidden layers (`antmaze`, `kitchen`, `visual-antmaze`) |
| | 3 hidden layers (`antsoccer`, `humanoidmaze`, `scene`, `cube-*`) |
| KL Coefficient ($\beta$) | 0.1 (`antmaze`, `humanoidmaze`, `kitchen`, `visual-antmaze`, `antsoccer`) |
| | 0.2 (`cube-*`, `scene`) |
| VAE Prior | state-conditioned isotropic Gaussian distribution over the latent |
| VAE Posterior | isotropic Gaussian distribution over the latent |
| Reconstruction Policy Decoder | isotropic Gaussian distribution over the action space |
| Latent Dimension of $z$ | 8 |
| Trajectory Segment Length ($H$) | 4 |
| Image Encoder Latent Dim | 50 |

Table 1: VAE training details.

In the online phase, our high-level policy is a Soft-Actor-Critic (SAC) agent (Haarnoja et al., 2018) with 10 critic networks, entropy backup disabled and LayerNorm added to the critics following the architecture design used in RLPD (Ball et al., 2023). Similar to **EXPLORE** (Li et al., 2024), we sample 128 offline samples and 128 online samples and add an RND reward bonus to all of the samples. While the original **EXPLORE** paper only added a reward bonus to the offline data, we add the reward bonus to both the offline data and online data, as it leads to better performance in goals where there is limited offline data coverage (see Appendix I.2).

## C. Implementation details for baselines

**SUPE**. We follow Li et al. (2024) (**EXPLORE**) to relabel offline data with optimistic reward estimates using RND and a reward model. For completeness, we describe the details below. We initialize two networks $g_\phi(s, z)$, $\bar{g}(s, z)$ that each outputs an 256-dimensional feature vector predicted from the state and (tanh-squashed) high-level action. During online learning, $\bar{g}(s, z)$ is fixed and we only update the parameters of the other network $g_\phi(s, z)$ to minimize the $L_2$ distance between the feature vectors predicted by the two networks on the new high-level transition $(s_0^{\text{new}}, z^{\text{new}}, r^{\text{new}}, s_H^{\text{new}})$:

$$\mathcal{L}(\phi) = \|g_\phi(s_0^{\text{new}}, z^{\text{new}}) - \bar{g}(s_0^{\text{new}}, z^{\text{new}})\|_2^2.$$

In addition to the two networks, we also learn a reward model $r_\zeta(s, z)$ that minimizes the reward loss below on the transitions $(s_0, z, r, s_H)$ from online replay buffer:

$$\mathcal{L}(\zeta) = \|r_\zeta(s_0, z) - r\|_2^2.$$

We then form an optimistic estimate of the reward value for the offline data as follows:

$$r_{\text{UCB}}(s, z) \leftarrow r_\zeta(s_0, z) + \alpha \| g_\phi(s_0, z) - \bar{g}(s_0, z) \|_2^2,$$

where $\alpha$ controls the strength of the exploration tendency (RND coefficient). For every environment with $-1/0$ rewards, we find that it is sufficient to use the minimum reward, $-1$, to label the offline data without a performance drop, so we opt for such a simpler design for our experiments. On `kitchen`, we use a reward prediction model since the reward is the number of tasks currently completed.

For online training batch from the replay buffer, we also add online reward bonus provided by RND with the same $\alpha$ coefficient. For all the baselines below, we also add the online RND bonus to the online batch to encourage exploration.

**ONLINE ONLY**. In the appendix, we additionally include a baseline that does not use offline data at all. This baseline directly runs the SAC agent from scratch online with no pretraining and no usage of the offline data.

**DBC+JSRL**. We use the diffusion model implementation from (Hansen-Estruch et al., 2023). Following the paper's implementation, we train the model for 3 million gradient steps with a dropout rate of 0.1 and a cosine decaying learning rate schedule from the learning rate of 0.0003. In the online phase, in the beginning of every episode, with probability $p$, we rollout the diffusion policy for a random number of steps that follows a geometric distribution $\text{Geom}(1-\gamma)$ before sampling actions from the online agent (inspired by (Li et al., 2023)). A RND bonus is also added to the online batch on the fly with a coefficient of 2.0 to further encourage online exploration of the SAC agent. The same coefficient is used in all other non-skill based baselines. For skill based baselines, we scale up the RND coefficient by the horizon length (4) to account for different reward scale. Following the **BC+JSRL** baseline used in **EXPLORE** (Li et al., 2024), we use a SAC agent with an ensemble of 10 critic networks, one actor network, with no entropy backup and LayerNorm in the critic networks. This configuration is used for all baselines on all environments. On `antmaze`, we perform a hyperparameter sweep on both $p = \{0.5, 0.75, 0.9\}$ and the geometric distribution parameter $\gamma = \{0.99, 0.995, 0.997\}$ on the large maze with the top right goal and find that $p = 0.9$ and $\gamma = 0.99$ works the best. We also use these parameters for the Visual `antmaze` experiments. On `antsoccer`, we use the same $p$ but raise the discount $\gamma$ to match the environment discount rate of 0.995. On `humanoidmaze`, we perform a sweep over $p = \{0.5, 0.75, 0.9\}$ on `humanoidmaze-medium-navigate` and find that $p = 0.75$ works best. We still use $\gamma = 0.995$. On Scene and Cube, we do a similar sweep for $p$ on the first task of Scene, Single Cube, and Double Cube, and find that $p = 0.5$ works best. We still use $\gamma = 0.995$. For `kitchen`, we perform a sweep on the parameter $p = \{0.2, 0.5, 0.75, 0.9\}$ and find that 0.75 works best. We use $\gamma = 0.99$. We take the minimum of one random critic for `antmaze`, `visual-antmaze`, `antsoccer`, and `humanoidmaze`, and the minimum of two random critics for `kitchen`, `scene`, and `cube`. For `antsoccer`, we performed a sweep on `antsoccer-arena-navigate` for each method over taking a minimum of 1 or 2 random critics, and found that taking a minimum of 1 critic was always better. For all methods, we use the same image encoder used in RLPD (Ball et al., 2023) for the `visual-antmaze` task, with a latent dimension of 50 (encoded image is a 50 dimensional vector), which is then concatenated with proprioceptive state observations.

**EXPLORE**. We directly use the official open-source implementation. The only difference we make is to adjust the RND coefficient from 1.0 to 2.0 and additionally add such bonus to the online replay buffer (the original method only adds to the offline data). Empirically, we find a slightly higher RND coefficient improves performance slightly. The SAC configuration is the same as that of the **DBC+JSRL** baseline. We found that taking the minimum of one random critic on `visual-antmaze` worked better for **EXPLORE** than taking the minimum of two, so all non-skill based baselines use this hyperparameter value.

**TRAJECTORY SKILLS**. This baseline is essentially our method but without using the trajectory encoder in the VAE to label trajectory segments (with high-level skill action labels), so all stated implementation decisions also apply to **SUPE**. Instead, we treat it directly as a high-level RL problem with the low-level skill policy completely frozen. The SAC agent is the same as the previous agents, except for on Visual AntMaze, where taking the minimum of 2 critics from the ensemble leads to better performance for **SUPE**, so we use this parameter setting for all skill-based benchmarks. We compute the high-level reward as the discounted sum of the rewards received every $H$ environment steps. During the $5 \times 10^3$ steps before the start of training, we sample random actions from the state-based prior. For Visual AntMaze, we use the learned image encoder from the VAE to initialize both the critic image encoder and the RND network. If using ICVF, we initialize the RND network and critic encoder with the ICVF encoder instead.

**HILP SKILLS**. This baseline is the same as the one above but with the skills from a recent unsupervised offline skill discovery method, HILP (Park et al., 2024b). We use the official open-source implementation from the authors and run the

pretraining to obtain the skill policies. Then, we freeze the skill policies and learn a high-level RL agent to select skills every $H$ steps.

**SUPE (HILP)**. This novel baseline is the same as **HILP SKILLS**, except that we also relabel the offline trajectories and use them as additional data for learning the high-level policy online (similar to our proposed method). To relabel trajectories with the estimated HILP skill, we compute the difference in the latent representation of the final state $s_H$ and initial state $s_0$ in the trajectory, so $\hat{z} \leftarrow \frac{\phi_{\text{HILP}}(s_H) - \phi_{\text{HILP}}(s_0)}{\|\phi_{\text{HILP}}(s_H) - \phi_{\text{HILP}}(s_0)\|_2}$. We normalize the skill vector since the pretrained HILP policies use a normalized vector as input. The high-level RL agent is the same as our method, except the skill relabeling is done using the latent difference rather than the trajectory encoder.

## D. Additional Related Work

**Options framework.** Many existing works on building hierarchical agents also adopt the options framework (Sutton et al., 1999; Menache et al., 2002; Chentanez et al., 2004; Mannor et al., 2004; Şimşek & Barto, 2004; Şimşek & Barto, 2007; Konidaris, 2011; Daniel et al., 2016; Srinivas et al., 2016; Fox et al., 2017; Bacon et al., 2017; Kim et al., 2019; Bagaria & Konidaris, 2019; Bagaria et al., 2024). The options framework provides a way to learn skills with varying time horizons, often defined by learnable initiation and/or termination conditions (Sutton et al., 1999). We use skills with a fixed horizon length because it allows us to avoid the additional complexity of learning initiation or termination conditions, and frame the skill pretraining phase as a simple supervised learning task.

**Reinforcement learning with observation-only offline data.** In this problem setting, the offline data only contains observations (e.g., videos) (Ma et al., 2022; Ghosh et al., 2023; Song et al., 2024). The closest to our work are Ma et al. (2022) and Ghosh et al. (2023), which focus on pre-training on observation-only offline data to extract good image representations that are suitable for down-stream RL tasks. Different from their work, we pre-train low-level skills rather than representations for online learning.

## E. Domain Details

**D4RL `antmaze` with Additional Goal Locations.** D4RL `antmaze` is a standard benchmark for offline-to-online RL (Fu et al., 2020; Ball et al., 2023) where an ant robot needs to navigate around a maze to a specified goal location. We benchmark on three mazes of increasing size, `antmaze-medium`, `antmaze-large`, and `antmaze-ultra`. We take the D4RL dataset for medium and large mazes as well as the dataset from Jiang et al. (2022) for the ultra maze (we use the `diverse` version of the datasets for all these layouts). We then remove the termination and reward information from the dataset such that the agent does not know about the goal location a priori. For each of the medium, large, ultra mazes, we test with four different goal locations that are hidden from the agent. See Figure 2a for a visualization of the mazes and the four goal locations that we use for each of them. We use a $-1/0$ sparse reward function where the agent receives $-1$ when it has not found the goal and it recieves $0$ when it reaches the goal location and the episode terminates. The ant always starts from the left bottom corner of the maze, and the goal for the RL agent is to reach the goal consistently from the start location. It is worth noting that the goal is not known a priori and the offline data is unlabeled. In order to learn to navigate to the goal consistently, the agent first needs to traverse in the maze to gather information about where the goal is located.

**D4RL `kitchen`** is another standard benchmark for offline-to-online RL (Fu et al., 2020; Nakamoto et al., 2024), where a Franka robot arm is controlled to interact with various objects in a simulated kitchen scenario. The desired goal is to complete four tasks (open the microwave, move the kettle, flip the light switch, and slide open the cabinet door) in sequence. In the process, the agent attains a reward equal to the number of currently solved tasks. The benchmark contains three datasets, `kitchen-mixed`, `kitchen-partial`, and `kitchen-complete`. `kitchen-complete` is the easiest dataset, which only has demonstrations of the four tasks completed in order. `kitchen-partial` adds additional tasks, but the four tasks are sometimes completed in sequence. `kitchen-mixed` is the hardest, where the four tasks are never completed in sequence. To adapt this benchmark in our work, we remove all the reward labels in the offline dataset.

We also include four additional domains repurposed from a goal-conditioned offline RL benchmark (Park et al., 2024a).

**OGBench `humanoidmaze`** is a navigation task similar to `antmaze`, but with the Ant agent replaced by a Humanoid agent. `humanoidmaze` is much more challenging than `antmaze` because Humanoid control is much harder with much higher action dimensionality (21 vs. 8). We benchmark on all six datasets in the benchmark. They involve three mazes of increasing size, `humanoid-medium`, `humanoid-large`, and `humanoid-giant`, and two types of datasets,

navigate (collected by noisy expert policy that randomly navigates around the maze) and stitch (containing only short segments that test the algorithm's ability to stitch them together).

**OGBench antsoccer** is a navigation task similar to antmaze, but with the added complexity where the Ant agent must first travel to the location of a soccer ball, then dribble the soccer ball to the goal location. We benchmark on the two navigate datasets, antsoccer-arena-navigate and antsoccer-medium-navigate. We only benchmark on task-1.

**OGBench cube-* and scene** are manipulation domains with a range of manipulation tasks for each domain. Cube domains involve using a robot arm to arrange blocks from an initial state to a goal state, which requires pick and place actions to move and/or stack blocks. We use both the cube-single and cube-double domains which include one and two blocks, respectively. We also benchmark on the Scene environment with two buttons, a drawer, and a window. The most difficult task requires the agent to perform 8 atomic actions: unlock the drawer and window, open drawer, pick up block, place block in drawer, close drawer, open window, lock drawer and window. We benchmark on all 5 tasks for cube-single, cube-double, and scene, and use the play datasets collected by non-Markovian expert policies with temporally correlated noise. Unlike the reward function used in the singletask versions of these tasks introduced after this paper was finished, we use a binary -1/0 reward on the completion of the entire episode to create a maximally sparse task. This is different from the reward in the recently introduced singletask versions of these tasks, where the reward is proportional to the number of subtasks completed (1 subtask for cube-single, 2 for cube-double, and 5 for scene).

Aside from the state-based domains above, we also consider a visual domain below to test the ability of our method in scaling up to high-dimensional image observations.

**visual-antmaze** is a benchmark introduced by Park et al. (2023a), where the agent must rely on $64 \times 64$ image observations of its surroundings, as well as proprioceptive information including body joint positions and velocities to navigate the maze. In particular, the image is the only way for the agent to locate itself within the maze, so successfully learning to extract location information from the image is necessary for successful navigation. The floor is colored such that any image can uniquely identify a position in the maze. The maze layout is the same as the large layout in the state-based D4RL antmaze benchmark above and we also use the same additional goals. The reward function and the termination condition are also the same as the state-based benchmark.

## F. ICVF Implementation Details and Ablation Experiments for **visual-antmaze**

We use the public implementation from the authors of (Ghosh et al., 2023) and run the ICVF training for 75000 gradient steps to obtain the pre-trained encoder weights, following (Li et al., 2024). Then, we initialize the encoder of the RND network with these weights before online learning. It is worth noting that this is slightly different from the prior work (Li et al., 2024) that initializes both the RND network and the critic network. In Figure 7, we examine the performance of **SUPE**, **TRAJECTORY SKILLS**, and **EXPLORE** with and without ICVF. Both of the baselines perform much better with the ICVF initialization, suggesting that ICVF might play an important role in providing more informative exploration signal. **SUPE**, without using ICVF, can already outperform the baselines with ICVF. By both extracting skills from offline data and training with offline data, we are able to learn better from less informative exploration signals. We also observe that initializing the critic with ICVF (as done in the original paper (Li et al., 2024)) helps improve the performance of **EXPLORE** some, but does not substantially change performance.

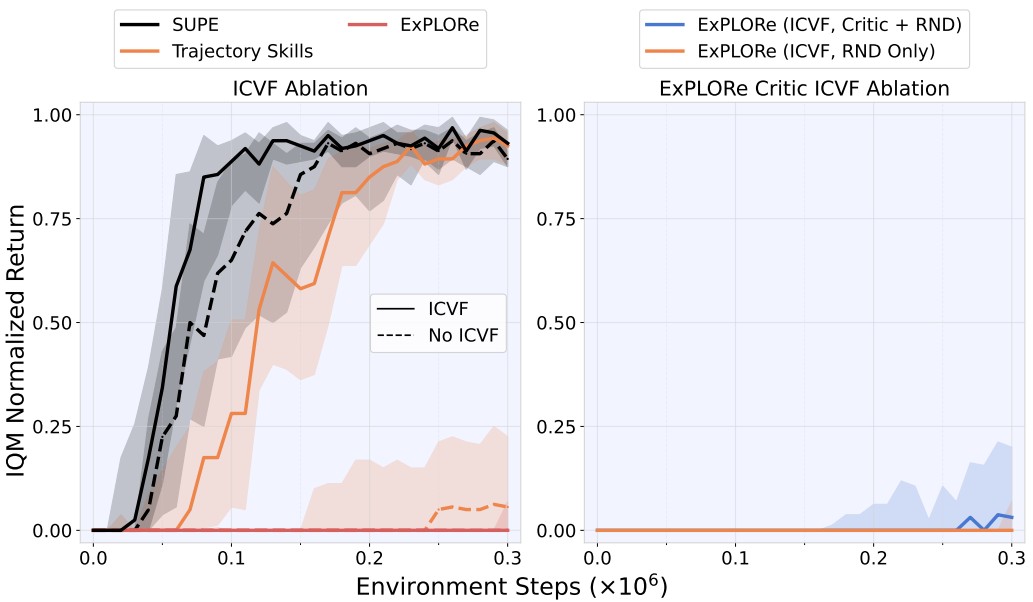

Figure 7: **Success rate on the `visual-antmaze` environment with and without ICVF. SUPE** works well without ICVF, almost matching the original performance. However, **Trajectory Skills** achieves far worse performance without ICVF, which shows that using offline data both for extracting skills and online learning leads to better utilization of noisy exploration bonuses. Initializing the critic with ICVF helps for **ExPLORe**, but does not substantially change performance.

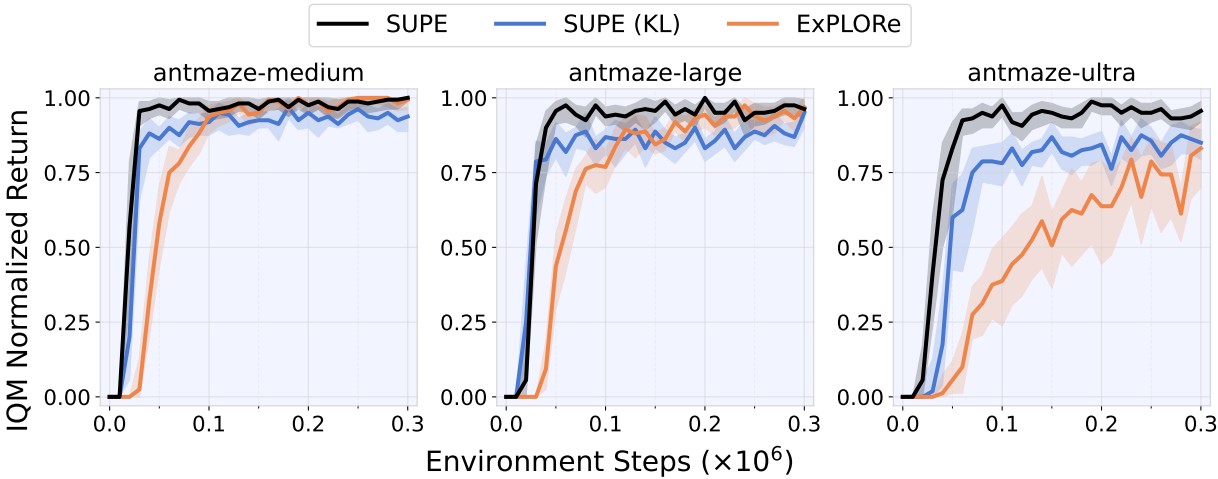

Figure 8: **IQM normalized return on three `antmaze` mazes, comparing SUPE with a KL regularized alternative (Ours (KL)).** We that **SUPE** consistently outperforms **Ours (KL)** on all three mazes, with initial learning that is at least as fast and significantly improved asymptotic performance. Only **SUPE** is able to meet or surpass the asymptotic performance of **ExPLORe** on all mazes.

## G. KL Penalty Ablation

In Figure 8, we compare the performance of **SUPE** with a version of our method that uses a KL-divergence penalty with the state-based prior (as used in a previous skill-based method (Pertsch et al., 2021)), **Ours (KL)**. In **SUPE**, as discussed in Section 4, we borrow the policy parameterization from (Haarnoja et al., 2018) and adopt a tanh policy parameterization with entropy regularization on the squashed space. Pertsch et al. (2021) parameterize the higher level policy as a normal distribution and is explicitly constrained to a learned state-dependent prior using a KL-divergence penalty, with a temperature parameter that is auto-tuned to match some target value by using dual gradient descent on the temperature parameter. They do not use entropy regularization. Keeping everything else about our method the same, we instantiate this alternative

policy parameterization in **Ours (KL)**. We sweep over possible target KL-divergence values $(5, 10, 20, 50)$ and initial values for the temperature parameter $(100, 1, 0.1)$ using the performance on `antmaze-large`, but find that these parameters do not substantially alter performance. As shown in Figure 8, **SUPE** performs at least as well as **Ours (KL)** in the initial learning phase, and has better asymptotic performance on all three mazes, matching or beating **EXPLORE** on all three mazes. It seems likely that not having entropy regularization makes it difficult to appropriately explore online, and that explicitly constraining to the prior may prevent further optimization of the policy. Attempts at combining an entropy bonus and KL-penalty lead to instability and difficulty tuning two separate temperature parameters. Additionally, in the Kitchen domain, the KL objective is unstable, since at some states the prior standard deviation is quite small, leading to numerical instability. In contrast, adopting the tanh policy parameterization from (Haarnoja et al., 2018) is simple, performs better, and encounters none of these issues in our experiments.

## H. Sensitivity Analysis

We picked one representative task from each domain: `antmaze-large-top-right`, `kitchen-mixed`, `humanoidmaze-giant-stitch`, `cube-single-task1`, `cube-double-task1`, `scene-task1`, and `antsoccer-arena-navigate`. Figure 9 shows the performance of our method with different RND coefficients. We see that an RND coefficent of zero leads to poor performance on `antmaze` and `humanoidmaze`, and that the best nonzero RND coeffcient varies between environments. For all experiments in the paper, we selected the middle value of eight, and kept it the same across all domains. Figure 10 shows the performance of our method with different skill horizon lengths. We see that while a horizon length of 2 attains better final performance on `antmaze-large` top right goal at the cost of slower initial exploration, it performs worse than a horizon length of 4 on all other environments. A longer horizon length of 8 seems to perform significantly better in `soccer` and `kitchen`, but performs the same or substantially worse in all other tasks. We found that using a horizon length of 4 generally works well on all tasks, so we used this length for all trajectory-skill based experiments in this paper.

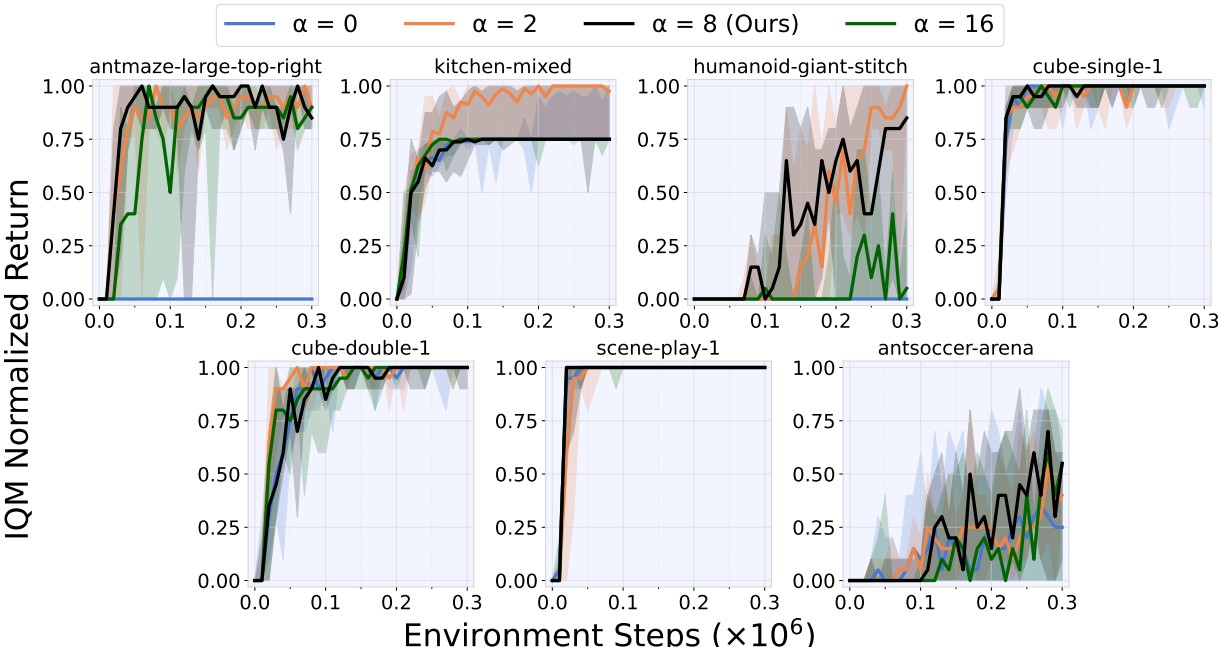

Figure 9: **Sensitivity analysis of the RND coefficient.** RND is essential to strong performance `antmaze` and `humanoidmaze`. The best coefficient varies between tasks.

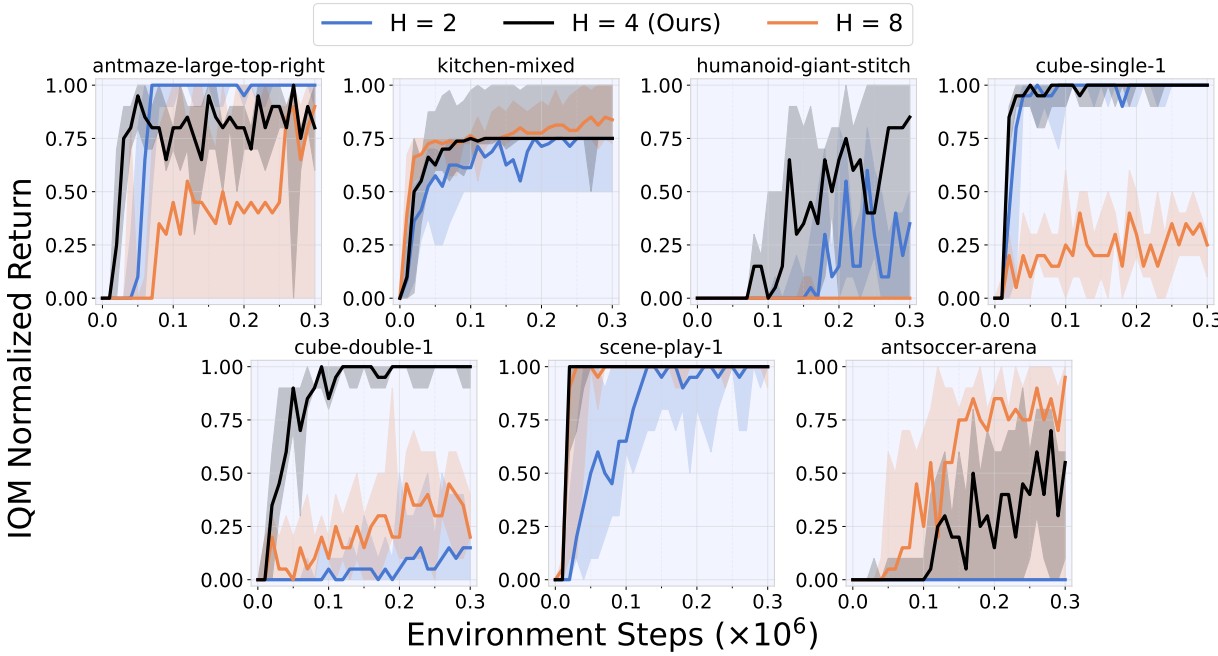

Figure 10: **Sensitivity analysis of the skill horizon length.** A horizon length of 4 generally performs the best across all environments. A horizon length of 2 performs relatively well in `antmaze-large`, but performs poorly in all other environments. A longer horizon length of 8 performs better than a horizon length of 4 in `soccer` and `kitchen`, but performs much worse in several other tasks.

## I. State-based D4RL Results

In this section, we summarize our experimental results on two state-based D4RL domains: `antmaze` and `kitchen`. Figure 11 shows the comparison of our method with all baselines.

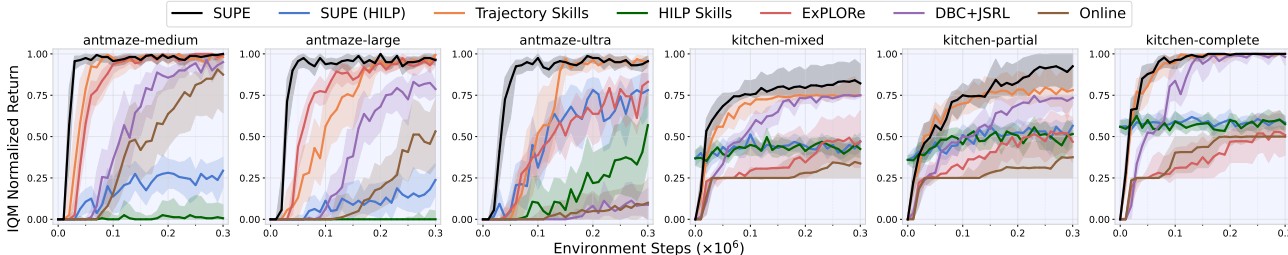

Figure 11: **IQM normalized return on individual `antmaze` and `kitchen` tasks. SUPE** achieves the strongest performance on all tasks. **TRAJECTORY SKILLS** learns much slower on all `antmaze` tasks. **EXPLORE** struggles to learn on `kitchen`, and performs worse as maze size increases. None of the other baselines are competitive on any tasks. Each curve is an average over four goals with 8 seeds for `antmaze`, and 16 seeds for `kitchen`.

### I.1. Exploration Efficiency

Figure 14 shows the percentage of the maze that the agent has covered throughout the training. The coverage of skill-based methods that do not use prior data during online learning, **TRAJECTORY SKILLS** and **HILP SKILLS**, significantly lags behind baselines that use offline data after 50000 environment steps. Many methods achieve similar coverage on `antmaze-medium`, likely because the maze is too small to differentiate the different methods. **SUPE** is able to achieve the highest coverage on the `antmaze-ultra`, and is only surpassed on `antmaze-large` by **SUPE (HILP)**, which has high first goal times and slow learning. Thus, the coverage difference can likely be at least partially attributed to **SUPE (HILP)** struggling to find the goal and continuing to explore after finding the goal. All non-skill based methods struggle to get competitive coverage levels on `antmaze-large` and `antmaze-ultra`. This suggests both pretraining skills and

the ability to leverage prior data online are crucial for efficient exploration, and our method effectively compounds their benefits.

### I.2. Full D4RL AntMaze Results

We evaluate the success rate of the our algorithm compared to the same baseline suite as in the main results section for each individual goal and maze layout and report the results in Figure 12. We also include **ExPLORe** both with and without an online RND bonus. Online RND helps **ExPLORe** the most for the `antmaze-medium` bottom-right goal, where there is sparse offline data coverage for a considerable radius around the goal. We hypothesize that with the absence of online RND, the agent is encouraged to only stay close to the offline dataset, making it more difficult to find goals in less well-covered regions. On the flip side, for some other goals with better offline data coverage, like the `antmaze-large` top-right goal, online RND can make the performance worse. For every goal location, **SUPE** consistently matches or outperforms all other methods throughout the training process.

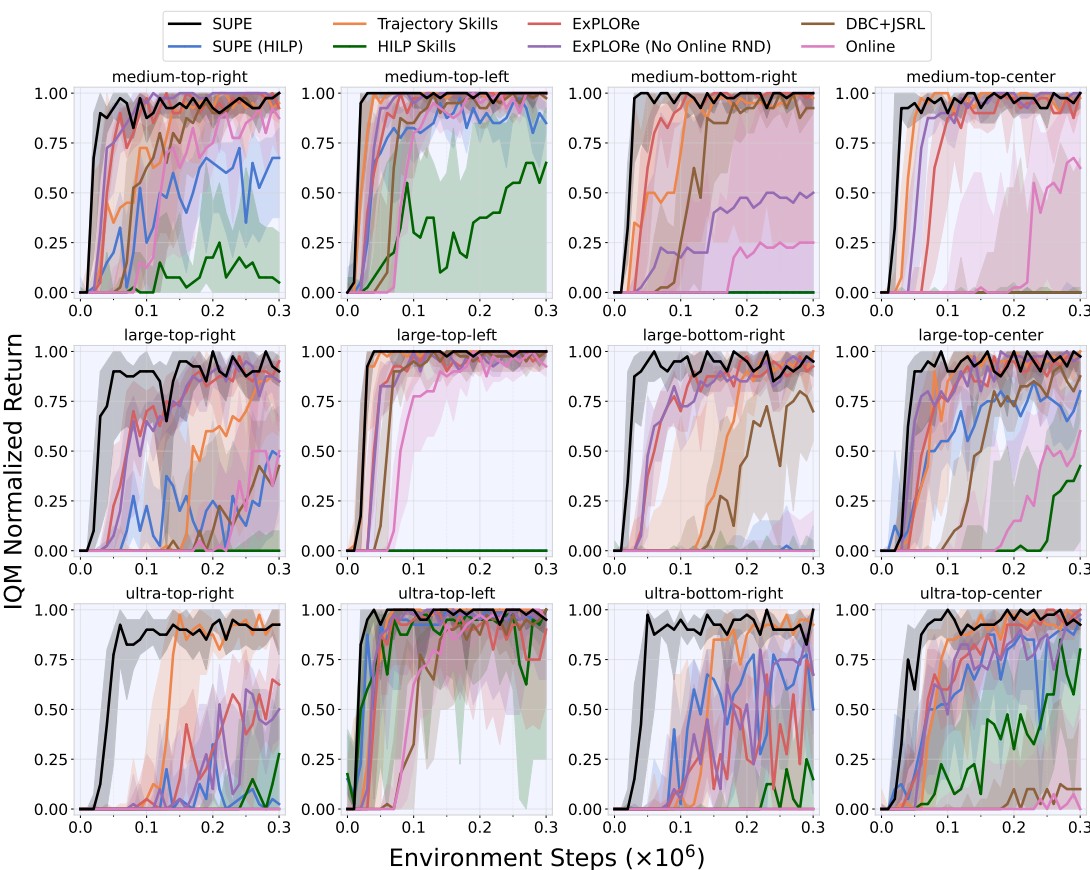

Figure 12: **IQM normalized return by goal location.** The addition of online RND in **ExPLORe** leads to better performance on goals with less offline data coverage, and slightly worse performance on goals well-represented in the dataset. **SUPE** consistently matches and outperforms all other methods on all goals throughout training.

We also evaluate the coverage at every goal location for every method for each maze layout and show the result in Figure 13. The coverage varies from goal location to goal location as some goal locations are harder to reach. Generally, the agent stops exploring once it has learned to reach the goal consistently. **SUPE** consistently has the best initial coverage for 11 out of 12 goals, though sometimes has lower coverage compared to other methods later in training. However, this is likely due in large part to successfully learning how to reach that goal quickly, and thus not exploring further.

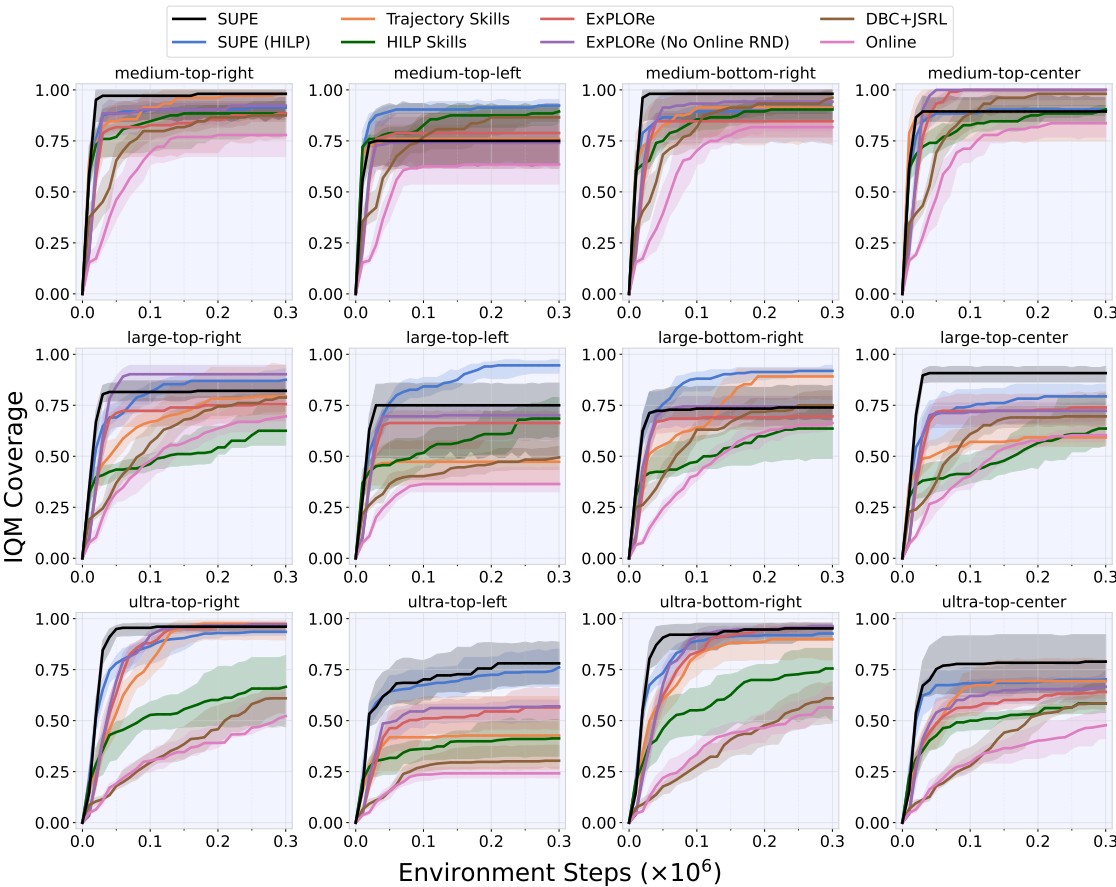

Figure 13: **IQM coverage for every goal location on three antmaze environments.** There is significant variation between goals, and **SUPE** consistently has the best initial coverage performance on 11 of 12 goals. Flattening coverage compared to other methods can be at least partially attributed to having already found the goal, and sucessfully optimizing reaching that goal, rather than continuing to explore after already finding the goal.

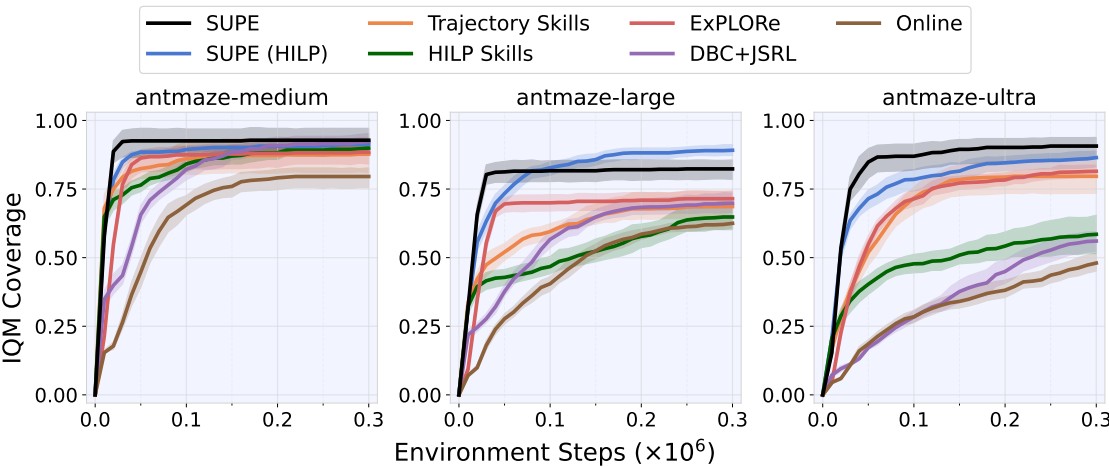

Figure 14: **IQM coverage on three different AntMaze mazes, averaged over runs on four goals. SUPE** has the best coverage performance on the challenging `antmaze-ultra`, and is only passed by **SUPE (HILP)** on `antmaze-large`. **TRAJECTORY SKILLS** and **HILP SKILLS** struggle to explore after initial learning, and **ONLINE ONLY** and **DBC+JSRL** generally perform poorly at all time steps.

| Maze Layout | Goal Location | Methods without Pretraining | | Methods with Pretraining | | | | |
| --- | --- | --- | --- | --- | --- | --- | --- | --- |
| | | ONLINE ONLY | ExPLORE | DBC+JSRL | TRAJECTORY SKILLS | HILP SKILLS | SUPE (HILP) | SUPE |
| medium | top-left | $71 \pm 5.0$ | $27 \pm 3.2$ | $60 \pm 8.1$ | $\mathbf{21 \pm 4.1}$ | $120 \pm 47$ | $27 \pm 6.2$ | $\mathbf{14 \pm 3.1}$ |
| | top-right | $100 \pm 16$ | $\mathbf{29 \pm 2.8}$ | $85 \pm 19$ | $76 \pm 26$ | $160 \pm 40$ | $\mathbf{72 \pm 36}$ | $\mathbf{22 \pm 3.2}$ |
| | bottom-right | $230 \pm 38$ | $35 \pm 4.9$ | $99 \pm 15$ | $77 \pm 34$ | $300 \pm 0$ | $270 \pm 33$ | $\mathbf{22 \pm 4.4}$ |
| | center | $210 \pm 32$ | $71 \pm 8.0$ | $260 \pm 28$ | $26 \pm 3.4$ | $260 \pm 28$ | $300 \pm 0.0$ | $\mathbf{18 \pm 1.7}$ |
| | *Aggregated* | $150 \pm 14$ | $40 \pm 2.0$ | $130 \pm 10$ | $50 \pm 11$ | $210 \pm 17$ | $170 \pm 13$ | $\mathbf{19 \pm 1.8}$ |
| large | top-left | $72 \pm 10$ | $33 \pm 2.9$ | $52 \pm 3.3$ | $\mathbf{22 \pm 4.2}$ | $300 \pm 0$ | $300 \pm 0.0$ | $\mathbf{21 \pm 2.8}$ |
| | top-right | $220 \pm 20$ | $49 \pm 7.7$ | $220 \pm 28$ | $190 \pm 27$ | $280 \pm 20$ | $110 \pm 36$ | $\mathbf{27 \pm 2.6}$ |
| | bottom-right | $280 \pm 15$ | $34 \pm 1.8$ | $160 \pm 22$ | $140 \pm 22$ | $280 \pm 21$ | $260 \pm 19$ | $\mathbf{21 \pm 1.8}$ |
| | top-center | $220 \pm 28$ | $\mathbf{48 \pm 5.2}$ | $120 \pm 8.8$ | $\mathbf{59 \pm 12}$ | $240 \pm 23$ | $\mathbf{33 \pm 8.5}$ | $39 \pm 6.2$ |
| | *Aggregated* | $200 \pm 8.9$ | $41 \pm 2.7$ | $140 \pm 13$ | $100 \pm 13$ | $270 \pm 12$ | $180 \pm 13$ | $\mathbf{27 \pm 1.7}$ |
| ultra | top-left | $76 \pm 7.0$ | $34 \pm 4.9$ | $91 \pm 11$ | $36 \pm 11$ | $\mathbf{39 \pm 21}$ | $\mathbf{15 \pm 5.3}$ | $\mathbf{17 \pm 3.6}$ |
| | top-right | $300 \pm 0.0$ | $92 \pm 20$ | $290 \pm 7.8$ | $120 \pm 14$ | $260 \pm 19$ | $150 \pm 32$ | $\mathbf{37 \pm 5.5}$ |
| | bottom-right | $300 \pm 0.0$ | $70 \pm 8.0$ | $300 \pm 0.0$ | $130 \pm 16$ | $240 \pm 28$ | $67 \pm 12$ | $\mathbf{34 \pm 6.0}$ |
| | top-center | $230 \pm 35$ | $29 \pm 5.5$ | $230 \pm 29$ | $\mathbf{75 \pm 16}$ | $100 \pm 32$ | $\mathbf{17 \pm 1.7}$ | $22 \pm 4.4$ |
| | *Aggregated* | $230 \pm 9.3$ | $56 \pm 5.4$ | $230 \pm 9.1$ | $90 \pm 7.1$ | $160 \pm 14$ | $61 \pm 8.2$ | $\mathbf{27 \pm 2.4}$ |
| **Aggregated** | | $190 \pm 6.9$ | $46 \pm 2.3$ | $160 \pm 6.3$ | $80 \pm 5.9$ | $210 \pm 11$ | $130 \pm 7.4$ | $\mathbf{25 \pm 1.4}$ |

Table 3: **The number of environment steps** ($\times 10^3$) **taken before the agent find the goal.** Lower is better. The first goal time is considered to be $300 \times 10^3$ steps if the agent never finds the goal. We see that our method is the most consistent, achieving performance **as good as or better than all other methods** in each of the 4 goals across 3 different maze layouts. The error quantity indicated is standard error over 8 seeds. The method that has the lowest mean is in bold and all the other methods with values that are not statistically significantly higher are also in bold. We use the one-sided $t$-test ($p = 0.05$) and bold all the methods that are not statistical significantly worse than the best method (the method that achieves the best mean performance).

### I.3. D4RL Play Dataset

Since there is limited performance difference between the diverse and the play datasets, we only report the performance on the diverse datasets. For completeness, we also include the results of the play datasets in Figure 15. The results on the play datasets are consistent with our results in the main body of the paper where our method outperforms all baseline approaches consistently with better sample efficiency.

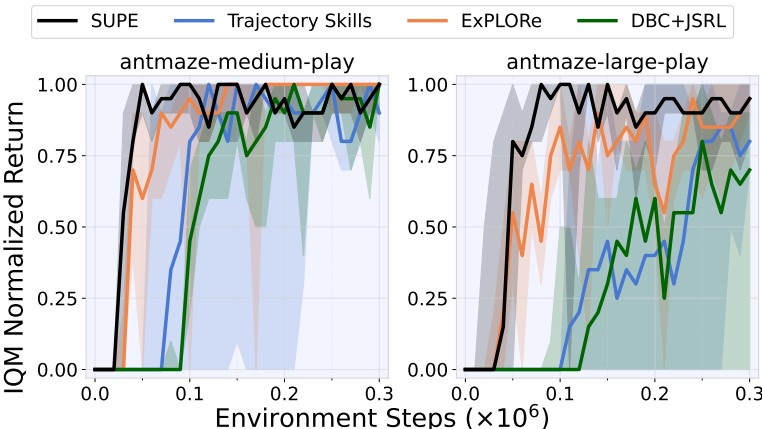

Figure 15: **IQM normalized return of our method on the play datasets. SUPE** outperforms all baselines, similar to the results on the diverse datasets (Figure 11). We average over 4 seeds.

## J. OGBench Results

In this section, we include the full results on individual tasks for each of the five OGBench domains (humanoidmaze, cube-single, cube-double, scene, and antsoccer) (Park et al., 2024a).

## J.1. HumanoidMaze

As shown in Figure 16, our method substantially outperforms all prior methods on the difficult `humanoidmaze` environment. It is the only method to achieve nonzero return on the more difficult `large` and `giant` mazes, and performs approximately four times better than the next best baseline, **TRAJECTORY SKILLS**, on the `medium` environments. These results show that on difficult, long horizon tasks, using offline data during online learning is essential for strong exploration performance.

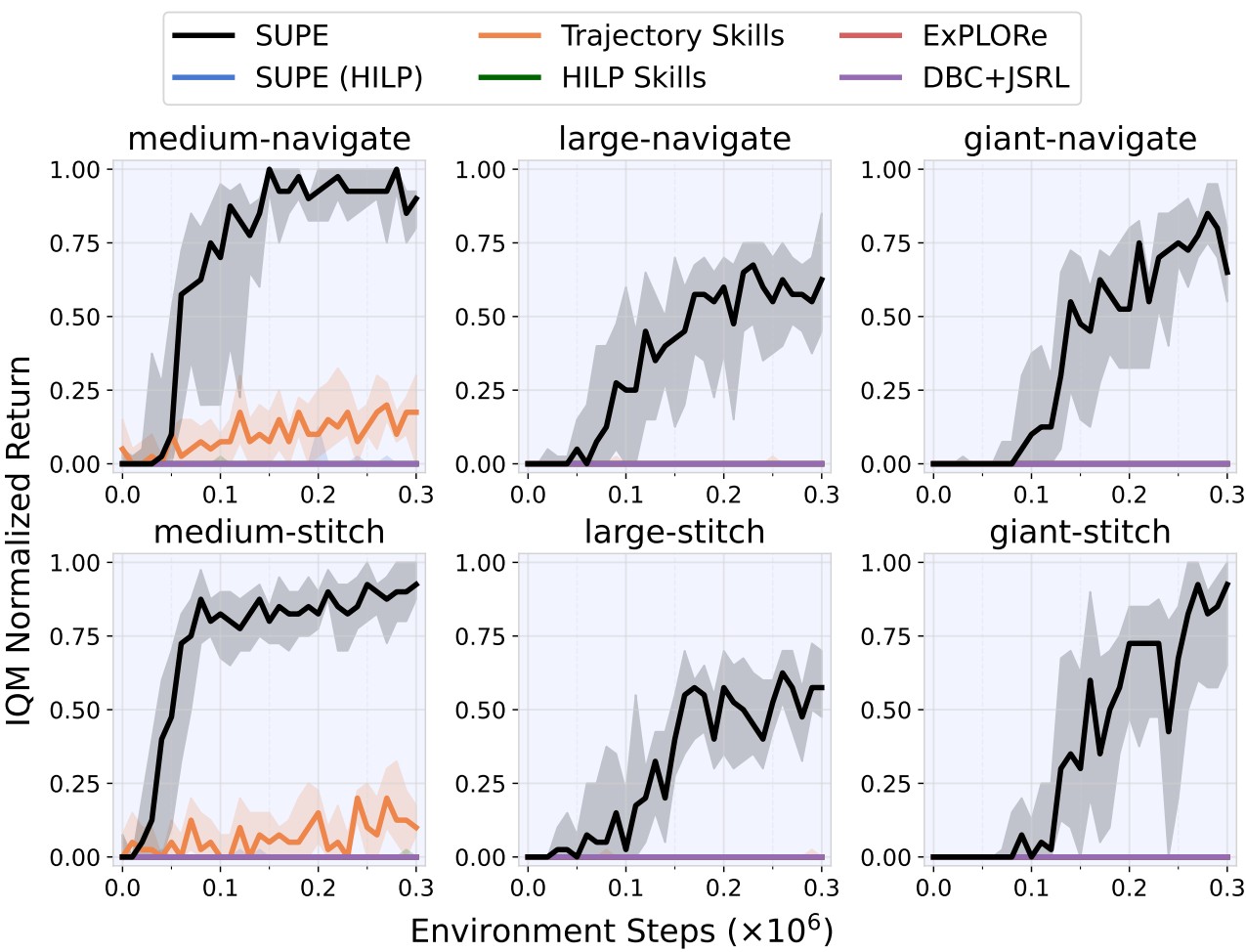

Figure 16: **IQM normalized return on six `humanoidmaze` tasks.** **SUPE** is the only method that obtain more than 0.5 IQM normalized return. All baselines either completely fail or only achieve less than 0.2 IQM normalized return on easier mazes (e.g., **TRAJECTORY SKILLS** on `medium-navigate` and `medium-stitch`). We average over 8 seeds.

## J.2. Cube and Scene

As shown in Figure 17, **SUPE** matches or outperforms the next best baseline on 12 of the 15 tasks. The novel baseline that we introduce **SUPE (HILP)** which also uses offline data for skill pretraining and online learning outperforms **SUPE** on three of the `scene` tasks, which further shows how using offline data twice is critical. Additionally, on one of the difficult `cube-double` manipulation tasks, **SUPE** is the only method to achieve nonzero reward. Non-skill based methods **EXPLORE** and **DBC+JSRL** performs reasonably well on the easier `cube-single` and `scene-task1`, but struggle to achieve significant return on the more difficult tasks, which shows how extracting structure from skills is critical for solving challenging tasks.

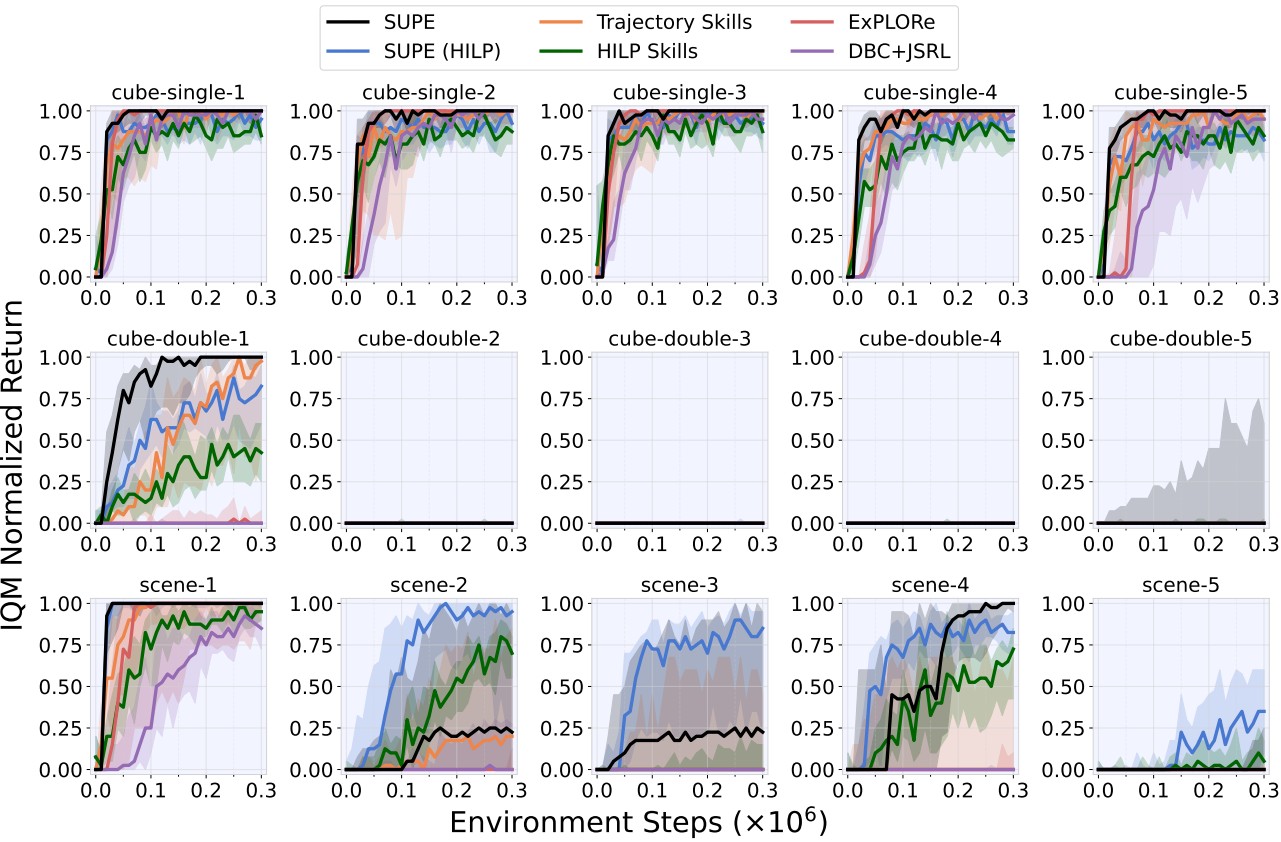

Figure 17: **IQM normalized return on individual tasks in `cube` and `scene` domains.** `cube` has 10 tasks in total (5 on `cube-single` and 5 on `cube-double`). `scene` has 5 tasks in total. We average over 8 seeds.

## J.3. AntSoccer

As shown in Figure 18, **SUPE (HILP)** outperforms all baselines on both `antsoccer-arena` and `antsoccer-medium`. We also see that on both tasks, **SUPE** and **SUPE (HILP)** outperform **TRAJECTORY SKILLS** and **HILP SKILLS**, which demonstrates the importance of using offline data twice to accelerate online exploration.

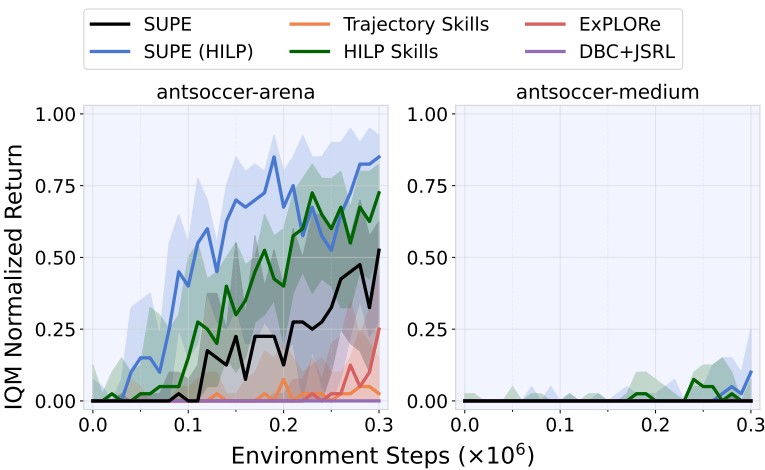

Figure 18: **IQM normalized return on individual tasks in the `antsoccer` domain.** We average over 8 seeds.

# K. Sensitivity to Offline Data Quality

To provide more insights on how the quality of the offline data affects the performance of our method, we perform additional analysis on the `antmaze-large` environment.

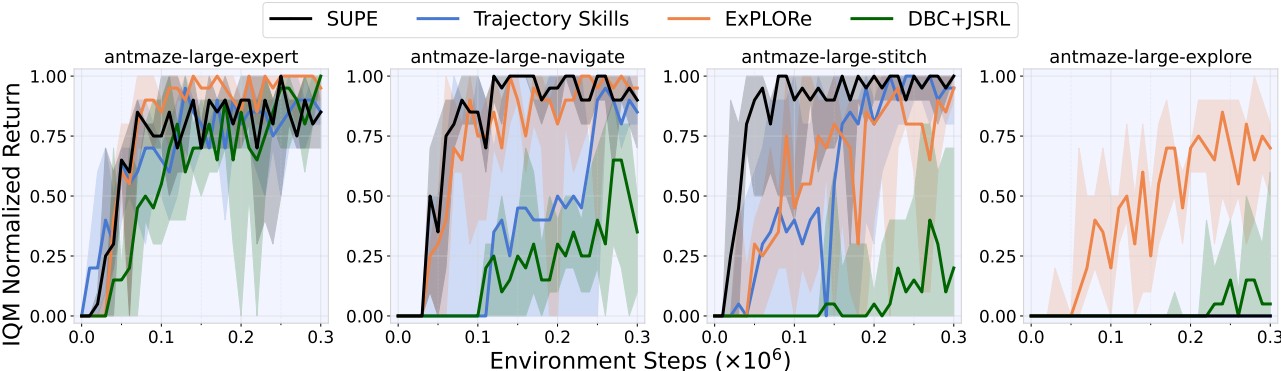

Figure 19: **Performance comparison on `antmaze-large-top-right` with different offline datasets.** *Expert*: collected by a non-noisy expert policy; *Navigate*: collected by a noisy expert policy that randomly navigates the maze from OGBench (Park et al., 2024a); *Stitch*: collected by the same noisy expert policy but with much shorter trajectory length (also from OGBench (Park et al., 2024a)); *Explore*: random exploratory trajectories collected by moving the ant in random directions re-sampled every 10 environment steps, with large action noise added (also from OGBench (Park et al., 2024a)). We average over 4 seeds.

## K.1. Expert Data to Random Exploratory Data

In Figure 19, we consider four additional offline datasets for the `antmaze-large` task with decreasing dataset quality:

1. **Expert**: collected by a non-noisy expert policy that we train ourselves.

2. **Navigate**: collected by a noisy expert policy that randomly navigates the maze (from OGBench (Park et al., 2024a)).

3. **Stitch**: collected by the same noisy expert policy but with much shorter trajectory length (from OGBench (Park et al., 2024a))

4. **Explore**: collected by moving the ant in random directions, where the direction is re-sampled every 10 environment steps. A large amount of action noise is also added (from OGBench (Park et al., 2024a)).

As expected, the baseline **EXPLORE** shows a gradual performance degradation from `Expert` to `Navigate`, to `Stitch`, and to `Explore`. All skill-based methods (including our method) fail completely on `Explore`. This is to be expected because the `Explore` dataset contains too much noise and the skills extracted from the dataset are likely very poor and meaningless. The high-level policy then would have trouble composing these bad skills to perform well in the environment. On `Navigate` and `Stitch`, our method matches or outperforms other baselines, especially on the more challenging `Stitch` dataset where it is essential to stitch shorter trajectory segments together. On `Expert`, **EXPLORE** performs slightly better than all other methods. We hypothesize that this is because with the expert data, online learning does not require as much exploration, and skill-based methods are mostly beneficial when there is a need for structured exploratory behaviors. Since the expert dataset has a very different distribution (much narrower) than the others, we performed a sweep over the KL coefficient in VAE training over {0.01, 0.05, 0.1, 0.2, 0.4, 0.8}, and found that 0.2 performed best, so we used these skills for both **SUPE** and **TRAJECTORY SKILLS** on this dataset.

## K.2. Data Corruptions

In this section, we study how robust our method is against offline dataset corruptions. We perform an ablation study on the `antmaze` domain on the large maze layout with two types of data corruption applied to the offline data:

1. *Insufficient Coverage*: All the transitions close to the goal (within a circle with a radius of 5) are removed.

2. *5% Data*: We subsample the dataset where only 5% of the trajectories are used for skill pretraining and online learning.

We report the performance on both settings in Figure 20. For the *Insufficient Coverage* setting, our method learns somewhat slower than the full data setting, but can still reach the same asymptotic performance, and outperforms or matches all baselines in the same data regime throughout the training process. For the *5% Data* setting, our method also reaches the same asymptotic performance as in the full data regime, and outperforms or matches all baselines throughout training. Overall, among the top performing methods in the `antmaze` domain, our method is the most robust, consistently outperforming the other baselines that either do not use pre-trained skills (**EXPLORE**) or do not use the offline data during online learning (**TRAJECTORY SKILLS**) in these data corruption settings.

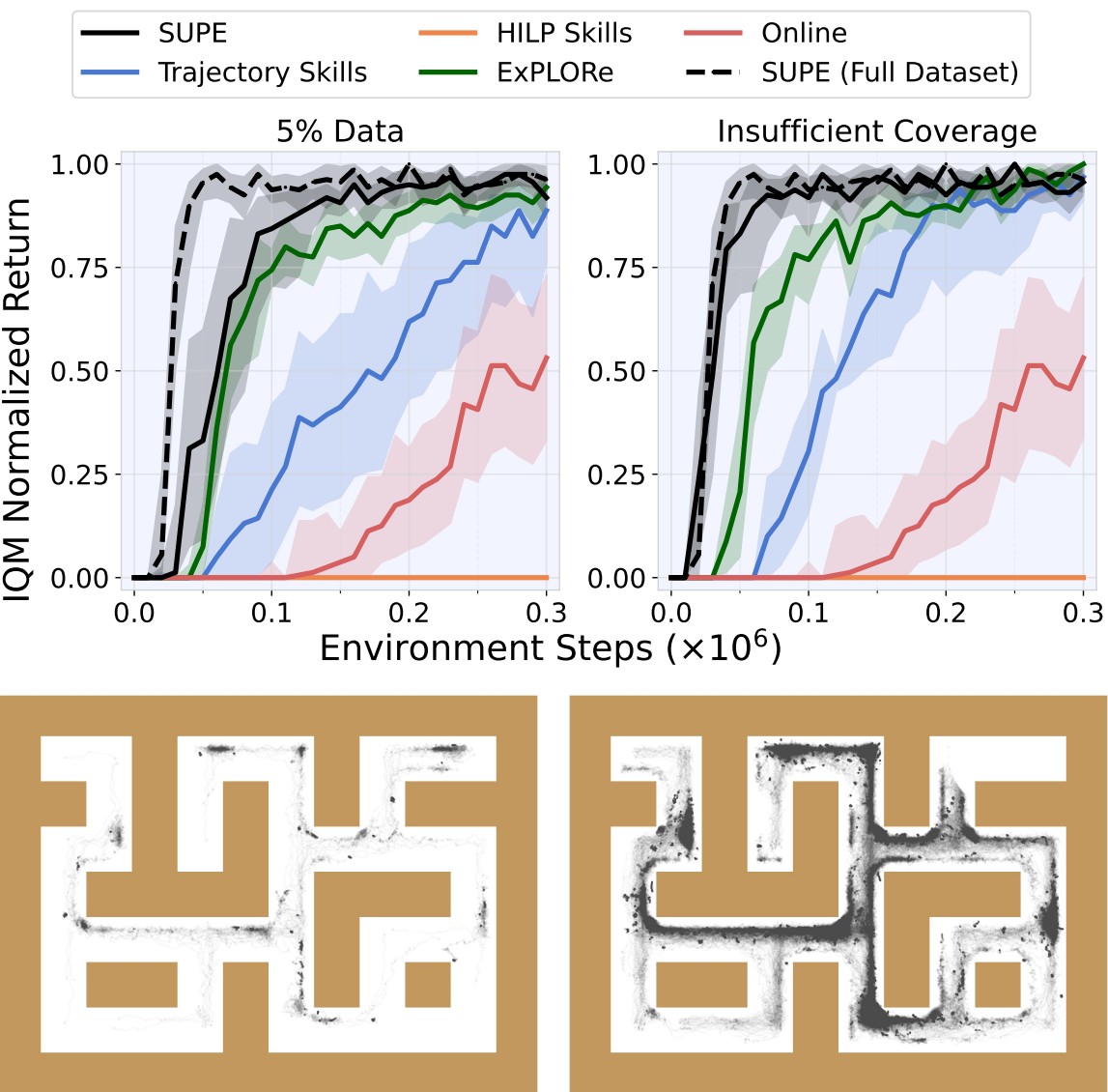

Figure 20: **Data corruption ablation on state-based `antmaze-large`.** *Top*: The success rate of different methods on these data corruption settings. *Bottom*: Visualization of the data distribution for each corruption setting. We experiment with two data corruption settings. Our method performs worse than the full data setting but still consistently outperforms all baselines. We average over 8 seeds.

