# OpenReview forum: "Leveraging Skills from Unlabeled Prior Data for Efficient Online Exploration"
_ICML.cc/2025/Conference — ICML 2025 poster_

### Official Review · Reviewer_rsMw · 2025-03-12

**Overall Recommendation:** 3

**Summary:**

In this paper, the authors propose SUPE, which combines the idea of pretraining offline skills and online exploration via hierarchical policy. SUPE first extracts low-level skills using a variational autoencoder, then pseudo-labels unlabeled trajectories with optimistic rewards and high-level action labels. Using this reward signal, SUPE train a high-level policy that composes pretrained skills for efficient exploration. In experiments, SUPE consistently outperforms previous methods across 42 long-horizon and sparse-reward tasks.

## Update after rebuttal
The reviewers addressed my questions and updated score accordingly.

**Claims And Evidence:**

The overall pipeline of using offline action primitives to train online HRL and applying UCB-style rewards is clear and convincing.

However, as noted below, there are similar ideas in the offline RL version, and the assumption that you have access to both (a) offline data and (b) an online environment can be a strong one.

The authors argue that "~ these methods cannot be directly used in our setting as they require offline data to have reward labels", but it remains unclear how common in the real world the special case of offline data being un-labeled but online interactions being rewarded is.

**Essential References Not Discussed:**

.

**Experimental Designs Or Analyses:**

The experimental design and analysis seem valid.

**Methods And Evaluation Criteria:**

The evaluation criteria for the proposed method seem appropriate, and experiments were conducted across a sufficient number of environments. Although the number of ablation studies is not very large, the two most important factors (the RND coefficient and skill horizon length) were well examined.

**Other Comments Or Suggestions:**

.

**Other Strengths And Weaknesses:**

The proposed method can be considered as an extension of OPAL (but online version), combined with a UCB approach. In other words, from a technical perspective, it seems equivalent to applying a UCB-style reward to OPAL and training it in an online setting.

**Questions For Authors:**

Why do the HILP-based methods in the kitchen environment of Figure 2 achieve a non-zero return right from the start?

What are the pros and cons of SUPE into HILP and the vanilla version, and what insights can you offer on which one should be chosen depending on the environment?

**Relation To Broader Scientific Literature:**

As the authors have claimed, there are not many clear methods that first learn offline action primitives and then utilize them in online settings. However, similar ideas do exist in the offline domain, and it might be better to provide a direct comparison with those approaches. (For more details, please refer to the section "Other Strengths and Weaknesses.")

**Theoretical Claims:**

.

---

> ### Author Rebuttal · Authors · 2025-04-01
>
> Thanks for your detailed review and insightful comments. We especially appreciate the comments on  the motivation. For your concern on the novelty of our method, we believe there might be a misunderstanding of our method – we believe that what we were describing is actually one of our baselines, “Trajectory Skills”. In our response, we explain in detail that we are not just a naive extension of OPAL combined with UCB (that is the baseline “Trajectory Skills”) and the differentiating step in our method makes a huge difference in performance from this naive extension as demonstrated in our experiments.
>
> **"The proposed method can be considered as an extension of OPAL (but online version), combined with a UCB approach. In other words, from a technical perspective, it seems equivalent to applying a UCB-style reward to OPAL and training it in an online setting."**
>
> We believe there is a misunderstanding. Our method is not equivalent to an extension of OPAL combined with a UCB approach. Rather, our “Trajectory Skills” baseline is equivalent to an online version of OPAL combined with a UCB approach from a technical perspective. Our “Trajectory Skills” baseline pretrains skills the same way as OPAL and learns a high-level online RL agent with a UCB approach to encourage exploration. The only, yet very important difference that distinguishes SUPE from this baseline is the pseudo-relabeling of the offline data that allows it to be used as additional off-policy data during online learning. Without the relabeling, we would not be able to leverage the offline data as additional off-policy data for online learning. We showed in our paper that using the offline data this way online results in a consistent/large performance boost over the “Trajectory Skills” baseline (Figure 2 and 3).
>
> **"It remains unclear how common in the real world the special case of offline data being un-labeled but online interactions being rewarded is."**
>
> In robotics, having access to task specific data can be expensive as it often requires human demonstration (e.g., through teleoperation or carefully scripted policy). It is often much easier to have access to a diverse dataset that is not directly related to the downstream task and use unsupervised RL algorithms to pretrain on this diverse dataset, and then fine-tune on a downstream task. This paradigm also started to gain popularity in RL in recent years [1, 2, 3, 4, 5]. In particular, O2O [1], FB [5] and ExPLORe [4] all study the setting where the offline data is unlabeled but the reward information is available in the online phase. O2O and FB assume the reward function is available right from the beginning of the online phase, whereas ExPLORe and our paper assumes the reward signal can only be obtained from online interactions. While the relevance of this problem setting merits discussion (which we will add to our paper), we believe that the number of prior works on this in just the last few years suggest that the topic is of interest to the ML community.
>
> [1] "Unsupervised-to-Online Reinforcement Learning"
>
> [2] "Fast Imitation via Behavior Foundation Models"
>
> [3] "Reinforcement learning from passive data via latent intentions"
>
> [4] "Accelerating exploration with unlabeled prior data"
>
> [5] "Learning one representation to optimize all rewards"
>
> **“Why do the HILP-based methods achieve a non-zero return right from the start?”**
>
> The data distribution for D4RL Kitchen is high quality, and BC methods can achieve some non-zero return [1]. Thus, since the HILP skills are effective at mimicking dataset behavior, sampling HILP skills can potentially achieve nonzero return prior to online exploration.
>
> [1] D4RL: Datasets for deep data-driven reinforcement learning. arXiv preprint arXiv:2004.07219, 2020.
>
> **"What are the pros and cons of SUPE into HILP and the vanilla version, and what insights can you offer on which one should be chosen depending on the environment?"**
>
> From current empirical observation, SUPE appears to be more stable during offline learning, and excels at long-horizon locomotion tasks (ie. $\texttt{visual-antmaze}$, $\texttt{antmaze}$, $\texttt{humanoidmaze}$). SUPE (HILP) is competitive for some manipulation tasks ($\texttt{antsoccer}$, $\texttt{scene}$), and achieves higher initial performance in some environments.
>
> We would like to thank you again for your constructive feedback and detailed reviews. Please let us know if you have any other concerns or questions. **If we have successfully addressed all your concerns, could you kindly raise your rating?**

---

> > ### Comment · Reviewer_rsMw · 2025-04-07
> >
> > Thank you for the authors' rebuttal.
> > I agree that this work has novelty compared to "OPAL combined with a UCB approach", and pseud-relabelling is the key idea that makes this work perform better than OPAL combined with a UCB approach.
> >
> > My concerns are resolved, and let me update my assessment.

---

> > > ### Author Response · Authors · 2025-04-07
> > >
> > > Thank you for raising the score, and again for your detailed review and insightful comments. We are really glad that your concerns have been addressed!

---

### Official Review · Reviewer_S75k · 2025-03-12

**Overall Recommendation:** 3

**Summary:**

The paper introduces hierarchical RL to train high-level policy to utilize the unlabel offline trajectories, and train reward estiamation, high-level action (skill) and extra offline data in online learning to guide policy learning. It claims to achieve efficient exploration performance in sparse reward tasks by pseudo-labeling each trajectory segment with an upper-confidence bound (UCB) reward.

**Claims And Evidence:**

The claim that the algorithm achieves efficient exploration and faster online learning is supported by experiments in state-based and pixel-based reinforcement learning (RL) environments, demonstrating higher normalized returns compared to baselines. This claim is backed by clear and convincing empirical evidence, including detailed performance tables and learning curves.

**Essential References Not Discussed:**

See the Relation To Broader Scientific Literature section.

**Experimental Designs Or Analyses:**

Although the paper lacks existing methods as it claims, it compares several online reinforcement learning (RL) methods, such as EXPLORE, DBC+JSRL, and a skill discovery approach like HILP. This section of the experiment appears robust, supported by statistical significance tests.

**Methods And Evaluation Criteria:**

Based on my understanding, the proposed method consists of two steps. First, a high-level policy is pretrained on unlabeled trajectory data using a VAE-based approach to learn diverse skills. Second, during online RL, a skill-conditioned policy collects data by interacting with the environment, combining offline data from the pretraining step with online data for training. The evaluation uses goal reaching task with normalized returns like D4RL to assess performance.

The inclusion of offline trajectories is reasonable for improving exploration efficiency. However, it is unclear why skill extraction is critical for online RL, as the algorithm uses offline data twice—first for skill extraction and second to augment training data—without justifying the marginal benefit of pseudo-labeling skills. Can you explain why skill extraction via pseudo-labeling provides a significant advantage over directly using offline data for training? A strong justification or a fair ablation study would strengthen the rationale as the title claimed.

**Other Comments Or Suggestions:**

The qualitative results and reported learning curves are clear and well-designed. However, it is unclear how the skill-conditioned policy guides goal-reaching tasks during evaluation. I suggest adding visualizations, such as those in Figure 4 of BeCL [Yang et al., 2023], to illustrate how the policy operates conditioned on different latent skills. A response with such visuals could enhance my confidence in the skill policy’s effectiveness.


[1] Behavior Contrastive Learning for Unsupervised Skill Discovery
Rushuai Yang, Chenjia Bai, Hongyi Guo, Siyuan Li, Bin Zhao, Zhen Wang, Peng Liu, Xuelong Li

**Other Strengths And Weaknesses:**

Strengths:
1. clear writing and easy to follow.

Weakness:
1. see discussion in the Methods And Evaluation Criteria section.

**Questions For Authors:**

1. Table 1 shows that the KL coefficient and GRU layers vary across environments rather than being consistent. Can you explain the rationale for these differences? A clear justification would strengthen the method’s design choices.

**Relation To Broader Scientific Literature:**

The topic in this paper quite aligns with some unsupervised reinforcement learning problems, which improves efficient exploration ablility and downstream task performance by addtional pretraining phase, some papers for reference. I believe that such method can be also adopt into online RL setting.

[1] URLB: Unsupervised Reinforcement Learning Benchmark
Michael Laskin, Denis Yarats, Hao Liu, Kimin Lee, Albert Zhan, Kevin Lu, Catherine Cang, Lerrel Pinto, Pieter Abbeel

[2] CIC: Contrastive Intrinsic Control for Unsupervised Skill Discovery
Michael Laskin, Hao Liu, Xue Bin Peng, Denis Yarats, Aravind Rajeswaran, Pieter Abbeel

[3] Constrained Ensemble Exploration for Unsupervised Skill Discovery
Chenjia Bai, Rushuai Yang, Qiaosheng Zhang, Kang Xu, Yi Chen, Ting Xiao, Xuelong Li

[4] METRA: Scalable Unsupervised RL with Metric-Aware Abstraction
Seohong Park, Oleh Rybkin, Sergey Levine

[5] Explore, Discover and Learn: Unsupervised Discovery of State-Covering Skills
Víctor Campos, Alexander Trott, Caiming Xiong, Richard Socher, Xavier Giro-i-Nieto, Jordi Torres

[6] Behavior Contrastive Learning for Unsupervised Skill Discovery
Rushuai Yang, Chenjia Bai, Hongyi Guo, Siyuan Li, Bin Zhao, Zhen Wang, Peng Liu, Xuelong Li

**Theoretical Claims:**

The paper does not present formal theoretical claims or proofs. It includes serveral equations defining the loss objective to optimize the proposed model’s performance. As such, there are no proofs to verify.

---

> ### Author Rebuttal · Authors · 2025-04-01
>
> Thanks for your detailed review and insightful comments. We especially appreciate the additional references you point out and the clarifying question on pseudo-labeling. For your question on how policy operates conditioned on different latent skills, we provide an additional detailed analysis that visualizes the skill latent and show how leveraging skills effectively reduces the exploration horizon, resulting in learning speedup online.
>
> **"I suggest adding visualizations, such as those in Figure 4 of BeCL [Yang et al., 2023], to illustrate how the policy operates conditioned on different latent skills"**
>
> We conducted additional analysis of the skill latent to demonstrate how skill discovery helps online exploration in one of our hardest domains, $\texttt{humanoidmaze}$. In particular, we randomly sample 16 latent vectors from our skill latent and roll-out our pre-trained low-level skill policy (corresponding to each of the sampled latent vectors) for an entire episode and visualize the x-y position throughout each of the 16 trajectories. For comparison, we also plotted the agent’s trajectories if actions were completely random. As shown in the figures [here](https://anonymous.4open.science/r/supe-rebuttal-8279/latent-viz/README.md), our unsupervised offline pretrained skills were able to navigate quite far from the initial positions even before any online learning. Such structured navigating behaviors allow the high-level policy to effectively operate at a reduced exploration horizon. Instead of training a low-level agent to predict $H$ correct actions in a row, the high-level policy only needs to predict 1 correct action to achieve a similar effect. As the exploration horizon is effectively reduced by a factor of $H$, we are able to train the high-level policy in SUPE to explore more efficiently and consequently solve the task much more quickly.
>
> **"Can you explain why skill extraction via pseudo-labeling provides a significant advantage over directly using offline data for training?"**
>
> The pseudo-labeling enables us to use offline data as additional off-policy data for the online training of the high-level agent. The high-level agent operates at a longer time-scale – taking only one high-level action ($z$) per 4 environment steps. Pseudo-labeling skills allows us to convert the low-level trajectories in the offline data $(s_t, a_t, s_{t+1}, a_{t+1}, \cdots)$ into high-level trajectories $(s_t, z_t, s_{t+H}, z_{t+H}, \cdots)$. Without pseudo-labeling, we would not be able to directly use the offline data to update the high-level RL agent online, which means that we could no longer use the offline data twice.
>
> **"Table 1 shows that the KL coefficient and GRU layers vary across environments rather than being consistent. Can you explain the rationale for these differences?"**
>
> The OGBench ($\texttt{humanoidmaze}$, $\texttt{antsoccer}$, $\texttt{scene}$, $\texttt{cube-single}$, $\texttt{cube-double}$) tasks are generally more difficult, so we use larger networks than for the simpler D4RL tasks. We select our network sizes for these OGBench environments according to the reference policy size used in the original OGBench paper [1]. For the manipulation tasks $\texttt{cube-*}$ and $\texttt{scene}$, the data distribution is narrower, and we found that a higher KL coefficient helps account for the difference in dataset diversity.
>
> [1] Park, Seohong, et al. "OGBench: Benchmarking offline goal-conditioned RL." arXiv preprint arXiv:2410.20092 (2024).
>
> **Related work references**
>
> Thanks for the additional references on unsupervised RL. We will incorporate them in our discussion in the paper and update in the camera ready version.
>
> We would like to thank you again for your constructive feedback and detailed reviews especially on the additional related work references. Please let us know if you have any other concerns or questions. **If we have successfully addressed all your concerns, could you kindly raise your rating?**

---

> > ### Comment · Reviewer_S75k · 2025-04-03
> >
> > Thank you for the authors’ efforts and their detailed response. Regarding my original question—"Does skill extraction via pseudo-labeling provide a significant advantage over directly using offline data for training?"—I understand your the clarification that extracting skills is necessary due to your algorithm’s design, which relies on an additional high-level policy to output latent features. However, I remain uncertain about whether it is truly necessary to extract skills and introduce an additional high-level policy (as opposed to using a single skill policy) for online exploration problem. Related works, such as EDL [1] and RLPD [2], demonstrate that directly leveraging offline data can effectively enhance online exploration performance, and these approaches appear to work well. I believe that it is more simple way to learn knowledge from prior data.
> >
> > Moreover, some works like METRA[3], and CIC[4] are still show promising exploration ability. Although it is orthogonal, they are just a low-level skill policy not relied on hierarchical guidance and skill feature is directly from simple uniform distribution. It is still unclear whether high-level policy for skill output is necessary. At least I haven't observe the obvious qualitative evidence about that. Based on these, I consider to maintain my score.
> >
> > [1] Explore, Discover and Learn: Unsupervised Discovery of State-Covering Skills. Víctor Campos, Alexander Trott, Caiming Xiong, Richard Socher, Xavier Giro-i-Nieto, Jordi Torres
> >
> > [2] Efficient Online Reinforcement Learning with Offline Data. Philip J. Ball, Laura Smith, Ilya Kostrikov, Sergey Levine
> >
> > [3] METRA: Scalable Unsupervised RL with Metric-Aware Abstraction Seohong Park, Oleh Rybkin, Sergey Levine
> >
> > [4] CIC: Contrastive Intrinsic Control for Unsupervised Skill Discovery Michael Laskin, Hao Liu, Xue Bin Peng, Denis Yarats, Aravind Rajeswaran, Pieter Abbeel

---

> > > ### Author Response · Authors · 2025-04-03
> > >
> > > Thanks for the promptly response and the follow-up concerns. For your concern that a simpler method like RLPD could work well without using skills, we have evidence in our paper that this is not the case. We show that we already had empirical results in our paper that compare our approach to a baseline that uses RLPD and our method is significantly more sample efficient (consistently across 8 domains). For your concern on existing low-level skill policy could do exploration well and hierarchical guidance is not necessary, we conducted additional experiments to show that just using low-level skill policy is not enough. Our experiment uses one of the skill methods that you pointed out, METRA, and showed that it is 100x slower at discovering goals online compared to our method on the $\texttt{antmaze}$ domain.
> > >
> > > > I remain uncertain about whether it is truly necessary to extract skills and introduce an additional high-level policy (as opposed to using a single skill policy) for online exploration problem. Related works, such as EDL [1] and RLPD [2], demonstrate that directly leveraging offline data can effectively enhance online exploration performance, and these approaches appear to work well. I believe that it is more simple way to learn knowledge from prior data.
> > >
> > > Thanks for raising the concern. Our baseline ExPLORe actually is using RLPD to directly leverage offline data (but with the added reward model and exploration bonus so that it can be applied in our setting where the offline data is unlabeled). In our experiments (Figure 2), we show that our method (SUPE) works consistently better than ExPLORe across all eight domains (by a large margin on the more challenging domains like $\texttt{humanoidmaze}$, $\texttt{visual-antmaze}$, $\texttt{scene}$, $\texttt{antsoccer}$). In particular, both skill-free methods (DBC+JSRL and ExPLORe) cannot achieve significant/competitive returns on any domain other than $\texttt{antmaze}$ and $\texttt{cube-single}$, and cannot solve the hardest environments ($\texttt{scene}$, $\texttt{cube-double}$, $\texttt{humanoidmaze}$) at all.
> > >
> > > >  Moreover, some works like METRA[3], and CIC[4] are still show promising exploration ability. Although it is orthogonal, they are just a low-level skill policy not relied on hierarchical guidance and skill feature is directly from simple uniform distribution.
> > >
> > > We conducted an additional experiment which shows that just using METRA objective for exploration is not enough for efficient exploration (100x worse than our method). For this experiment, we evaluated how quickly METRA is able to find the reward signal (reach the desired goal) online compared to our method. In the table below, we report the # of env steps that METRA took before the learned skills could reach the desired goal from the initial state (6 seeds). Our method took significantly fewer environment steps (on the order of 1/100x) compared to METRA. These results suggest that the $\texttt{antmaze}$ domain poses a significant exploration challenge, and naively using a diversity-encouraging objective (METRA) for online exploration can be extremely sample inefficient (100x worse than our method).
> > >
> > > The table below reports the average number of environment steps each method takes to reach the goal for the first time (all numbers are in million (x10^6) steps)
> > > | Layout and Goal Location | SUPE | METRA   | METRA (goal reached rate)|
> > > |----------------------|------------|---------|-----------|
> > > | antmaze-medium-top-left      |   0.014+-0.0031       | 7.04  | 6/6
> > > | antmaze-medium-top-right     |   0.022+-0.0032       | 11.46 | 4/6
> > > | antmaze-medium-bottom-right  |   0.022+-0.0044        | inf     | 0/6
> > > | antmaze-medium-top-center    |   0.018+-0.0017         | inf     | 0/6
> > > | antmaze-large-top-left       |   0.021+-0.0028          | 5.8  | 6/6
> > > | antmaze-large-top-right      |   0.027+-0.0026         | 12.48 |  1/6
> > > | antmaze-large-bottom-right   |   0.021+-0.0018         | inf     | 0/6
> > > | antmaze-large-center         |   0.039+-0.0062          | 14.72 | 3/6
> > > | antmaze-ultra-top-left       |   0.017+-0.0036          | 6.72  | 6/6
> > > | antmaze-ultra-top-right      |   0.037+-0.0055          | inf     | 0/6
> > > | antmaze-ultra-bottom-right   |   0.034+-0.006         | inf     | 0/6
> > > | antmaze-ultra-top-center     |   0.022+-0.0044         | 24.48 | 1/6
> > >
> > > (for METRA, many runs never managed to find the goal before a fixed budget of environment steps ran out [24M for medium and large, 48M for ultra]. We only average over the seeds where the goal is reached. We also report the number of seeds where goals were successfully reached at least once before the budget ran out. For SUPE, all goals were reached at least once for all eight seeds.)
> > >
> > > Thank you again for your constructive feedback and promptly response to our rebuttal. Please let us know if you have any other concerns or questions. **If we have successfully addressed all your concerns, could you kindly raise your rating?**

---

### Official Review · Reviewer_D8KW · 2025-03-13

**Overall Recommendation:** 3

**Summary:**

The paper presents a new algorithm on combining offline data and online RL where one can use the offline data to learn skills, and online perform exploration in the state space and skill space jointly, and relabel offline data with the exploration bonus. The paper performs comparisons with a few baselines on several tasks, and demonstrated that the sample efficiency of the proposed algorithm is superior to the previous baselines.

**Claims And Evidence:**

The claim that the proposed algorithm performs better than previous baselines is backed by evidence on a wide range of tasks.

**Essential References Not Discussed:**

I think the paper overall does a great job on the literature review on skill discovery and general offline to online rl, but I think it is missing the most relevant literatures on the o2o RL with unlabeled offline RL works (I am not sure each paper will provide a meanful baseline for this paper, but would be great to compare if any of them apply here), just to list a few:

Sharma, Archit, Rehaan Ahmad, and Chelsea Finn. "A state-distribution matching approach to non-episodic reinforcement learning." arXiv preprint arXiv:2205.05212 (2022).

Ma, Yecheng Jason, et al. "Vip: Towards universal visual reward and representation via value-implicit pre-training." arXiv preprint arXiv:2210.00030 (2022).

Ghosh, Dibya, Chethan Anand Bhateja, and Sergey Levine. "Reinforcement learning from passive data via latent intentions." International Conference on Machine Learning. PMLR, 2023.

Song, Yuda, Drew Bagnell, and Aarti Singh. "Hybrid Reinforcement Learning from Offline Observation Alone." International Conference on Machine Learning. PMLR, 2024.

**Experimental Designs Or Analyses:**

While the experiment demonstrates evidence of the strong empirical performance of the algorithm, one might question the source of improvement comparing to previous baselines. As the paper already compared with the baselines which also performs active exploration with offline data, it seems like the key ingredient is the skill discovery. However, a more careful analysis of how learning the skill helps the exploration or learning is lacking. The paper will benefit a lot from such zoomed-in analysis on the benefit and understanding the role of skill discovery.

Also intuitively, the method seems to heavily depend on the coverage/diversity of the offline data. I believe the paper will also benefit from conducting an analysis on offline datasets with different coverage and discover the failure mode of the algorithm.

It would also be interesting to see the performance gap between using the proposed algorithm, but offline data actually has rewards labeled.

**Methods And Evaluation Criteria:**

Yes.

**Other Comments Or Suggestions:**

see above

**Other Strengths And Weaknesses:**

Overall I think the paper is not really providing any new technique, but the performance is strong enough to make some contribution. However, the experiment should not only include performance comparison, but also conduct more detailed analysis.

**Questions For Authors:**

one minor point is on the usage of the term ucb: I do not see reference to ucb in the paper, is the bonus used in the paper just a general exploration bonus (but not really ucb)?

**Relation To Broader Scientific Literature:**

I believe the paper provides a strong baseline on the offline to online with unlabeled offline data problem, which is beneficial to the community.

**Theoretical Claims:**

N/A

---

> ### Author Rebuttal · Authors · 2025-04-01
>
> Thanks for your detailed review and insightful comments. We especially appreciate the additional references you point out and the clarifying question on UCB. For your concern on the dependence on quality of the offline data, we had ablation studies in our appendix that demonstrate the robustness of our approach on offline datasets with different levels of quality. For your question on how skill discovery helps, we provide an additional detailed analysis that visualizes the skill latent and show how leveraging skills effectively reduces the exploration horizon, resulting in a learning speedup online.
>
> **Analysis on offline datasets with different coverage and failure modes**
>
> We had ablation studies on datasets with different levels of quality in the Appendix J and showed our approach is robust against different dataset corruptions. A notable failure mode is the $\texttt{explore}$ dataset where the dataset contains completely random actions, and where the skill-pretraining methods we use in SUPE cannot learn meaningful behaviors. Aside from this failure case, SUPE is able to learn efficiently from offline datasets with limited coverage (e.g., missing transitions around the goal location, Figure 19, right), limited scale (e.g., 95% of trajectories removed, Figure 19, left), and suboptimal data (e.g., the $\texttt{stitch}$ dataset, Figure 18). We will refer to these ablation studies and discuss the failure modes of our approach in the main paper in the camera ready version.
>
> **Detailed analysis on skill discovery**
>
> We conducted additional analysis of the skill latent to demonstrate how skill discovery helps online exploration in one of our hardest domains, $\texttt{humanoidmaze}$. In particular, we randomly sample 16 latent vectors from our skill latent and roll-out the corresponding low-level skill policy for an entire episode and visualize the x-y position throughout each of the 16 trajectories. For comparison, we also plotted the agent’s trajectories if actions were completely random. The graphs can be viewed [here](https://anonymous.4open.science/r/supe-rebuttal-8279/latent-viz/README.md).  We can see that fixed skill latents are able to navigate quite far from the starting point. Such structured navigating behaviors allow the high-level policy to effectively operate at a reduced exploration horizon. Instead of training a low-level agent to predict $H$ correct actions in a row, the high-level policy only needs to predict 1 correct action to achieve a similar effect. As the exploration horizon is effectively reduced by a factor of $H$, we are able to train the high-level policy in SUPE to explore more efficiently and solve the task much faster.
>
> **Additional references**
>
> Thanks for the additional references. In our paper, we actually do have experiments using ICVF (Ghosh et al., 2023) on the high-dimensional image domain in Appendix E. ICVF is used complementary to our proposed method and our baselines. In our experiments (Figure 6), we found that our method works well without ICVF, achieving a much better performance than our baseline “Trajectory Skills” with or without the ICVF pre-trained representations. We also found that using ICVF representations further improves our method but only marginally. Even with ICVF, many of our baselines still failed to achieve significant return (e.g., ExPLORe as shown in Figure 6, right).
>
> Conceptually, both VIP (Ma et al., 2022) and ICVF are very related but different from our work in two ways: 1) they focus on pre-training on observation-only offline data whereas we assume access to actions, 2) they focus on extracting good image representations from the offline pre-training, whereas we pretrain low-level skills and retain the offline data to be used as additional off-policy transitions for online learning.
>
> Both (Sharma et al., 2022) and (Song et al., 2024) are also relevant and we will cite and discuss them (along with the two above) in the camera ready version of our paper.
>
> **Question about UCB**
>
> In the RL literature, UCB is typically used on the optimal value function with a confidence interval (e.g., [1]). We take the same definition of the term UCB in our paper and to describe the upper-confidence bound $r_{\mathrm{UCB}}$ of a reward function – with high probability the reward value is at most $r_{\mathrm{UCB}}$. As what we stated in our method section, we followed prior work (Li et al., 2024) to instantiate this estimate of the bound using an RND network and a reward model. The reward model predicts the mean of the reward estimate and the RND provides an estimate on how wide the confidence interval is.
>
> [1] "Minimax regret bounds for reinforcement learning."
>
> We would like to thank you again for your constructive feedback and detailed reviews especially on the related work references. Please let us know if you have any other concerns or questions. **If we have successfully addressed all your concerns, could you kindly raise your rating?**

---

> > ### Comment · Reviewer_D8KW · 2025-04-04
> >
> > I appreciate the authors for their rebuttal. My major concerns are addressed (I understand due to time limit the result is only on one domain but for the final version it would be nice to have a more comprehensive version of the analysis). I will raise my score accordingly.

---

> > > ### Author Response · Authors · 2025-04-04
> > >
> > > Thank you for raising the score, and again for your constructive feedback and detailed reviews. We are really glad that your concerns have been addressed. We will add a more comprehensive version of the analysis in our final version of the paper.

---

### Official Review · Reviewer_2HE7 · 2025-03-14

**Overall Recommendation:** 4

**Summary:**

The paper studies how to leverage skills pre-trained by unsupervised RL on unlabeled data to improve exploration while solving downstream tasks online. The proposed method (SUPE) first pre-trains skills via a trajectory VAE, then labels offline data with optimistic rewards, and finally trains a policy mapping states into skill vectors while interacting with the environment. Experimentally, SUPE is shown to outperform existing baselines and several challenging continuous control tasks.

## Update after rebuttal

The rebuttal clarified all my doubts, and I think the new experiments will provide good additional value to the paper. I am thus voting for acceptance.

**Claims And Evidence:**

The main claim that SUPE enables more efficient exploration and faster learning at downstream is supported by a good empirical evidence on several challenging exploration problems. Maybe one issue is that SUPE is compared with very few baselines: only two are existing algorithms, while the others are variants of the proposed method. While I guess this can be explained by the fact that there exist very few methods for efficient exploration in this context, I still wonder if other existing techniques could be adapted to provide further evidence about the complexity of the considered problems and the efficiency of SUPE. For instance, could "diversity-encouraging" objectives of recent unsupervised skill discovery methods (like DIAYN or METRA) be used as a regularizer to incentivize exploration during the online learning phase?

**Essential References Not Discussed:**

None

**Experimental Designs Or Analyses:**

As far as I understand from the details given in the paper, the experimental protocol is sound and the results are convincing.

**Methods And Evaluation Criteria:**

The proposed method makes sense. It is simple and intuitive. However, I think it has some limitations:
1. It seems quite incremental since it combines many existing techniques (although in a smart and non-trivial way)
2. The idea of re-using pre-training data during the online exploration phase, while effective as shown experimentally, does not seem very practical. Now skills are pre-trained on relatively small datasets, but imagine you want to scale this up to the pre-training settings of, e.g., recent large foundation models (say, with hundreds of terabytes of data). We cannot expect that these data can be deployed and moved around together with the pre-trained skills to be reused for each new task we encounter. I think the approach would be much more convincing and practical if it found a way to "compress" relevant knowledge to drive exploration at downstream in a learned model rather than keeping the full pre-training dataset
3. The way skills are pre-trained, essentially through behavioral cloning, together with the experimental results make me wonder whether this approach works only on very high quality data (e.g., expert demonstrations), and possibly that covers well the space that will have to be explored at downstream. If this is the case, I think it is another significant limitation, especially because the exist skill pretraining methods (like HILP) that can make use of "high-coverage" data (like EXORL) which does not contain any useful behavior, but from which useful behaviors can still be extracted

The evaluation protocol makes sense. The considered tasks are sufficiently diverse and challenging to showcase exploration capabilities. Again, the only limitation may be about the pre-training data: if it is given by high-qualitity demonstrations that cover well the space to be explored at test time, than SUPE's performance would not be very surprising

**Other Comments Or Suggestions:**

None

**Other Strengths And Weaknesses:**

Strengths
- the paper studies a significant problem
- the paper is well written
- the proposed approach is simple and effective

Weaknesses (see above for details)
- the proposed approach is a bit incremental
- need of high-quality pretrainig data
- need to retain the pretraining data for the adaptation phase

**Questions For Authors:**

1. What's the difference between this setting and the recently proposed "Unsupervised-to-Online Reinforcement Learning" (Kim et al., 2024)? Only that the reward function here is not given but rather only observed through samples from the environment?
2. If this is the case, could you motivate in what real problems this is useful? For instance, in robotics most of the time the reward is chosen by the designer, so it is not unreasonable to assume that the function itself is given at downstream instead of being "discovered" through interaction

**Relation To Broader Scientific Literature:**

They relate to the broader field of unsupervised pre-training followed by task-specific adaptation

**Theoretical Claims:**

None

---

> ### Author Rebuttal · Authors · 2025-04-01
>
> Thanks for your detailed review and insightful comments. We especially appreciate your clarifying question on the setting of our work. Regarding your concern on not comparing to “diversity-encouraging” objectives, we conducted additional experiments with METRA and showed that METRA finds a reward signal online much slower than our method. In addition, our experiments already involved a comparison to a variant of our method that leverages “diversity-encouraging” objectives in the offline pre-training phase (SUPE (HILP)). For your concern about the dependence on offline data quality, our ablation studies in the appendix of our original submission demonstrated the robustness of our approach to offline datasets with different levels of quality, quantity, and coverage.
>
> **Experiments with “diversity-encouraging” methods**
>
> We conducted additional experiments to evaluate how fast a diversity-encourage objective (METRA) is able to find the reward signal (reach the desired goal) online. In the table below, we report the # of env steps that METRA took before the learned skills could reach the desired goal from the initial state (6 seeds). Our method took significantly fewer environment steps (on the order of 1/100x) compared to METRA. These results suggest that the $\texttt{antmaze}$ domain poses a significant exploration challenge, and naively using a diversity-encouraging objective (METRA) for online exploration can be extremely sample inefficient (100x worse than our method). The table can be viewed [here](https://anonymous.4open.science/r/supe-rebuttal-8279/README.md).
>
> These results may not be too surprising, since METRA does not leverage offline data. One may wonder what might happen if we adapt METRA to use offline data. This is actually quite similar to SUPE (HILP), a variant of our method which uses a “diversity-encouraging” objective for offline pretraining instead of a behavioral cloning objective. Similar to METRA, HILP learns a metric space of states that preserves temporal distance. SUPE (HILP) is otherwise identical to SUPE (e.g., uses the offline data for both skill pretraining and high-level agent learning online) with the only difference being that the offline skill pretraining objective is different. SUPE works better across six of the eight domains we experimented on, and is competitive with SUPE (HILP) on the other two.
>
> Our method is not tied to a specific choice of skill pre-training method or online exploration bonus. We used VAE behavioral cloning skill pretraining and an RND exploration bonus because they are simple and effective in the benchmark tasks considered.
>
> **Does SUPE only work on very high quality data?**
>
> In Appendix J of our submission, we had some ablation studies on datasets with different levels of quality. In Figure 18&19, we show that our approach is reasonably robust against various dataset corruptions (e.g., insufficient coverage around the goal location, noisy experts with short trajectory lengths). In addition, our approach can be easily adapted to work in conjunction with different skill pre-training methods. On datasets that are diverse and “high-coverage”, we can leverage pre-trained skills that are more suitable for these properties (e.g., HILP) [1]. As mentioned in the previous section, one variant of our approach (SUPE (HILP)) leverages such skills.
>
> [1] "Foundation policies with hilbert representations."
>
> **“What's the difference between this setting and (Kim et al., 2024)? Could you motivate in what real problems this is useful?”**
>
> Your understanding is correct. In the O2O paper, the reward function is available at the start of the online phase. Our setting is harder since we don’t have access to the reward function and the reward signals must be obtained through environment interactions.
>
> In robotics, manually specifying a good reward function is challenging and it often suffers from misspecification issues (when the optimal policy of the specified reward misaligns with the expected optimal policy). For example, if you would like to specify a reward function for a home robot to clean a house, there are a lot of things to consider: should the robot clean the kitchen? What if the robot bumps into a wall? How do we define if cleanliness is acceptable? A much more appealing interaction with the robot would be that the robot first does some exploratory cleaning behavior (informed by its prior experience, possibly from its prior cleaning experience in another house) and then the user gives feedback (sparse reward signal) to the robot on whether the cleaning is done properly in a safe manner, and the robot improves upon the signal provided. We believe our work is an important first step towards making this part of our reality in the future.
>
> Thank you again for your constructive feedback and detailed reviews. Please let us know if you have any other concerns or questions. **If we have successfully addressed all your concerns, could you kindly raise your rating?**

---

> > ### Comment · Reviewer_2HE7 · 2025-04-04
> >
> > Thanks for the detailed response and for running additional experiments. My concerns have been nicely addressed. I am increasing my score.

---

> > > ### Author Response · Authors · 2025-04-05
> > >
> > > Thank you for raising the score, and again for your detailed reviews and insightful comments. We are really glad that your concerns have been addressed!

---

### Decision · Program_Chairs · 2025-05-01

**Decision:**

Accept (poster)

**Comment:**

The paper introduces SUPE, an approach that combines offline skill pretraining with online reinforcement learning (RL) to improve exploration. SUPE achieves faster learning and more efficient exploration compared to existing methods on multiple long-horizon and sparse-reward tasks.

The main strengths of the paper lie in its empirical results and novel use of offline data for online exploration. There are concerns initially, but the authors' response addresses most concerns. All reviewers agree to accept this manuscript.